# Unveiling RNA structure-mediated regulations of RNA stability in wheat

Haidan Wu[1,4], Haopeng Yu [1,2,4], Yueying Zhang [1,2,4], Bibo Yang[2,4], Wenqing Sun[1], Lanying Ren[1], Yuchen Li[1], Qianqian Li[2,3], Bao Liu [1]✉, Yiliang Ding [2]✉ & Huakun Zhang [1]✉

Despite the critical role of mRNA stability in post-transcriptional gene regulation, research on this topic in wheat, a vital agricultural crop, remains unclear. Our study investigated the mRNA decay landscape of durum wheat (*Triticum turgidum* L. ssp. *durum*, BBAA), revealing subgenomic asymmetry in mRNA stability and its impact on steady-state mRNA abundance. Our findings indicate that the 3' UTR structure and homoeolog preference for RNA structural motifs can influence mRNA stability, leading to subgenomic RNA decay imbalance. Furthermore, single-nucleotide variations (SNVs) selected for RNA structural motifs during domestication can cause variations in subgenomic mRNA stability and subsequent changes in steady-state expression levels. Our research on the transcriptome stability of polyploid wheat highlights the regulatory role of non-coding region structures in mRNA stability, and how domestication shaped RNA structure, altering subgenomic mRNA stability. These results illustrate the importance of RNA structure-mediated post-transcriptional gene regulation in wheat and pave the way for its potential use in crop improvement.

The stability of mRNA is a critical factor in regulating post-transcriptional activity, significantly influencing gene expression. Wheat plays a vital role in global nutrition, providing around one-fifth of the world's calorie and protein intake, emphasizing its imperative contribution to diets worldwide. Recent advances in deep sequencing have supplied extensive wheat genome data, facilitating functional genomics exploration[1,2]. More recently, an increased focus on wheat chromatin stability, exemplified by techniques such as ATAC-seq and HiC-seq, has aimed to comprehend transcriptional activity[3–5]. However, understanding the mRNA stability of the wheat transcriptome in regulating post-transcriptional activity remains elusive.

Notably, wheat is a prime example of an allopolyploid plant species that was characterized by hybridization followed by whole genome duplication[6,7]. In allopolyploids, a subgenome refers to one of the distinct sets of chromosomes that originates from one of the contributing species. The allopolyploidization process often induces significant genomic changes due to the merging of divergent genomes, leading to both structural and functional incompatibilities[8]. One common feature of these genomic changes is subgenome-biased expression, which undergoes dynamic selection during wheat domestication and environmental adaptation[1,2,9]. Recent studies have highlighted a transcriptional imbalance, with approximately 30% of wheat homoeolog triads, genes from the A, B, and D subgenomes of hexaploid wheat, exhibiting asymmetries in steady-state RNA abundances[10]. Given that steady-state RNA abundance primarily relies on the balance between RNA transcription and decay rates, we propose that RNA decay likely plays a role in the observed asymmetries in steady-state RNA abundances[11,12].

[1]Key Laboratory of Molecular Epigenetics of the Ministry of Education, Northeast Normal University, Changchun, China. [2]Department of Cell and Developmental Biology, John Innes Centre, Norwich Research Park, Norwich, UK. [3]Guangdong Provincial Key Laboratory of Applied Botany & Key Laboratory of South China Agricultural Plant Molecular Analysis and Genetic Improvement, South China Botanical Garden, Chinese Academy of Sciences, Guangzhou, China. [4]These authors contributed equally: Haidan Wu, Haopeng Yu, Yueying Zhang, Bibo Yang. ✉e-mail: baoliu@nenu.edu.cn; yiliang.ding@jic.ac.uk; zhanghk045@nenu.edu.cn

In various species, multiple mRNA features have been reported to affect RNA decay. For instance, in *E. coli*, shorter mRNAs tend to exhibit greater stability[13]. Conversely, in Zebrafish, lengthening the 3′ UTR can suppress codon-mediated deadenylation, resulting in enhanced mRNA stability[14]. Apart from the 3′ UTR length, sequence content has been suggested to be important in affecting mRNA stability. For instance, the most well-studied sequence elements are AU-rich elements (AREs)[15]. However, elucidating the precise features within individual AREs that control stability remains challenging[16]. Additionally, codon optimality plays an important role in mRNA stability[17,18]. For instance, in yeast, unstable mRNAs tend to contain a higher proportion of non-optimal codons[17–20]. RNA structure is another crucial determinant of mRNA stability[21,22]. Genome-wide studies in humans and yeast have revealed that strong RNA structures in 3′ UTRs can prevent mRNA degradation[23–25]. However, certain decay factors, such as STAU1 in humans, often recognize stem-loop RNA structures[26]. In *Arabidopsis* and rice, in vivo RNA structure profiling methods using chemical probing with a reverse transcription stalling assay revealed that mRNAs with weak RNA structures in their 3′ UTRs tend to be less stable[27,28]. Conversely, mRNAs with strong RNA structures in their 3′ UTRs are more stable[27,28]. Another type of RNA structure profiling method using a reverse transcription mutation assay, showed opposite relationships between the 3′ UTR RNA structure and RNA decay rate[29]. Given these diverse associative measurements in plants, systematically characterizing mRNA stability-associated RNA structural motifs poses a considerable challenge.

In this study, we successfully generated the first wheat mRNA decay landscape and revealed that mRNA decay asymmetry is significantly associated with differential gene expression between subgenomes. Through comprehensive analysis, we explored the impact of various mRNA features on mRNA stability, highlighting RNA structures within the 3′ UTR as a prominent determinant in wheat. We systematically identified stability-associated RNA structural motifs across the wheat transcriptome, elucidating subgenomic preferences. Furthermore, we found that single-nucleotide variations (SNVs) selected during domestication could alter stability-associated RNA structural motifs, subsequently affecting subgenomic mRNA stability and resulting in alterations in subgenomic expression levels. The demonstrated examples validated that both natural and ethyl methanesulfonate (EMS)-induced mutations perturb the stability-associated RNA structural motifs, which are capable of tuning mRNA stability. Taken altogether, our findings provide valuable insights into the stability of the wheat transcriptome and the potential to utilize RNA structure for regulating gene expression in crop improvement.

## Results

### The wheat mRNA decay landscape revealed a broad range of decay rates

To investigate mRNA stability in wheat, we conducted genome-wide mRNA decay analyses in the tetraploid durum wheat cultivar, Kronos ($2n = 4\times = 28$, BBAA). To measure RNA decay, we treated 4-day-old seedlings with a transcription inhibitor (cordycepin)[30] and collected samples at a series of time points (Fig. 1a). At each time point, RNA was prepared for library construction and subjected to deep sequencing. We acquired ~55 million 150-nt paired-end reads with three biological replicates per set (Supplementary Data 1), which showed high reproducibility (mean Pearson correlation coefficient = 0.98 between replicates) (Supplementary Fig. 1). We further estimated the mRNA decay rates for over 33,430 mRNAs using maximum likelihood modeling[30] (Supplementary Data 2). The transcription of mRNAs in cordycepin-treated replicates was inhibited over time, as shown by a progressive decline in mRNA abundance (Fig. 1b). A fast decrease indicates a high decay rate, while a slow decrease suggests a low decay rate. Hierarchical clustering of mRNA abundance over time showed a diverse range of decay rates, from dark blue (slow decay rates) transitioning to

light blue (faster decay rates) on the heatmap (Fig. 1b). Even within the same gene family, genes can display varying decay rates. For instance, two genes from the *SWEET* (*Sugars Will Eventually be Exported Transporter*) family demonstrated distinct RNA decay rates, with *TaSWEET13e* showing a slow decay rate, resulting in a comparatively higher expression level than *TaSWEET17e*, which exhibited a rapid decay rate, yielding a lower expression level (Fig. 1c). Similar to other species[30,31], mRNA half-lives in the wheat transcriptome varied greatly, from the shortest half-life of 1.98 min to >24 h, with a mean of 6.15 h (Supplementary Data 2 and Fig. 1d). We performed gene ontology (GO) analysis for 10% of the genes with the fastest and slowest decay rates, respectively (Supplementary Fig. 2). We found that genes exhibiting the fastest decay rates are predominantly involved in processes such as response to auxin, signal transduction and ion transport. In contrast, genes with the slowest decay rates were primarily associated with translation, photosynthesis, and metabolic processes.

Among all mRNAs, both intronless genes and miRNA-targeted mRNAs have been reported to be less stable than other mRNAs in other species[31,32]. We assessed these two specific types of mRNAs in our wheat RNA decay libraries, by plotting their average RNA decay curves (Fig. 1e and Supplementary Data 3). We found that both intronless genes and miRNA-targeted mRNAs have comparatively higher decay rates than the overall average mRNA decay rate, (Fig. 1e). The average half-lives of intronless genes and miRNA-targeted mRNAs are 3.53 h and 5.28 h, respectively (Supplementary Fig. 3a, b). Notably, ~50% of both intronless genes and miRNA-targeted mRNAs have half-lives of less than 1 h (Supplementary Fig. 4). As RNA steady-state levels result from the balance between transcription and decay[33], we then examined the correlation between steady-state mRNA abundances and our mRNA decay rates. We found that mRNA decay rates significantly anti-correlate with steady-state mRNA abundance, indicating that mRNA stability contributes to steady-state mRNA abundance (Fig. 1f, $r = -0.28$, $p < 2.2e{-}16$, two-sided Pearson correlation test). Moreover, intronless genes and miRNA target genes showed stronger negative correlations with steady-state mRNA abundance (Fig. 1g, h). We further classified the genes into different biological functions where we found that genes involved in functions such as growth regulation, response to auxin, and protein disulfide oxidoreductase activity had much stronger negative correlations between mRNA stability and abundance (Fig. 1i). These results highlight the importance of RNA stability in regulating gene expression levels.

### Wheat subgenomes display asymmetry in mRNA decay, reflecting differences in steady-state mRNA abundance between subgenomes

Previous studies have suggested that wheat subgenomes display both transcriptional and translational asymmetry, implying neo- or subfunctionalization of wheat homoeologs, which is a key evolutionary feature[10,34]. Due to the significant contribution of RNA stability to steady-state mRNA abundance that we observed (Fig. 1f), we asked whether RNA stability also displayed an asymmetrical pattern, contributing to the subgenomic differences in steady-state mRNA abundances. To test this, we compared the RNA decay rates of homoeologs between the two subgenomes in tetraploid Kronos. We discovered that 3184 homoeologous gene pairs demonstrated significantly higher decay rates in the A subgenome compared to the B subgenome (Fig. 2a, example illustrated in b). In addition, 3414 homoeologous gene pairs had significantly lower decay rates in the A subgenome compared to the B subgenome (Fig. 2a, example illustrated in c). Furthermore, we assessed whether differential subgenomic RNA stability might be a contributing factor to steady-state mRNA abundance asymmetry. Indeed, we found significant positive correlations between the differential mRNA decay rates and the differential steady-state mRNA abundances for homoeologous gene pairs between A and B subgenomes. For gene pairs with higher decay rates in A over B, the

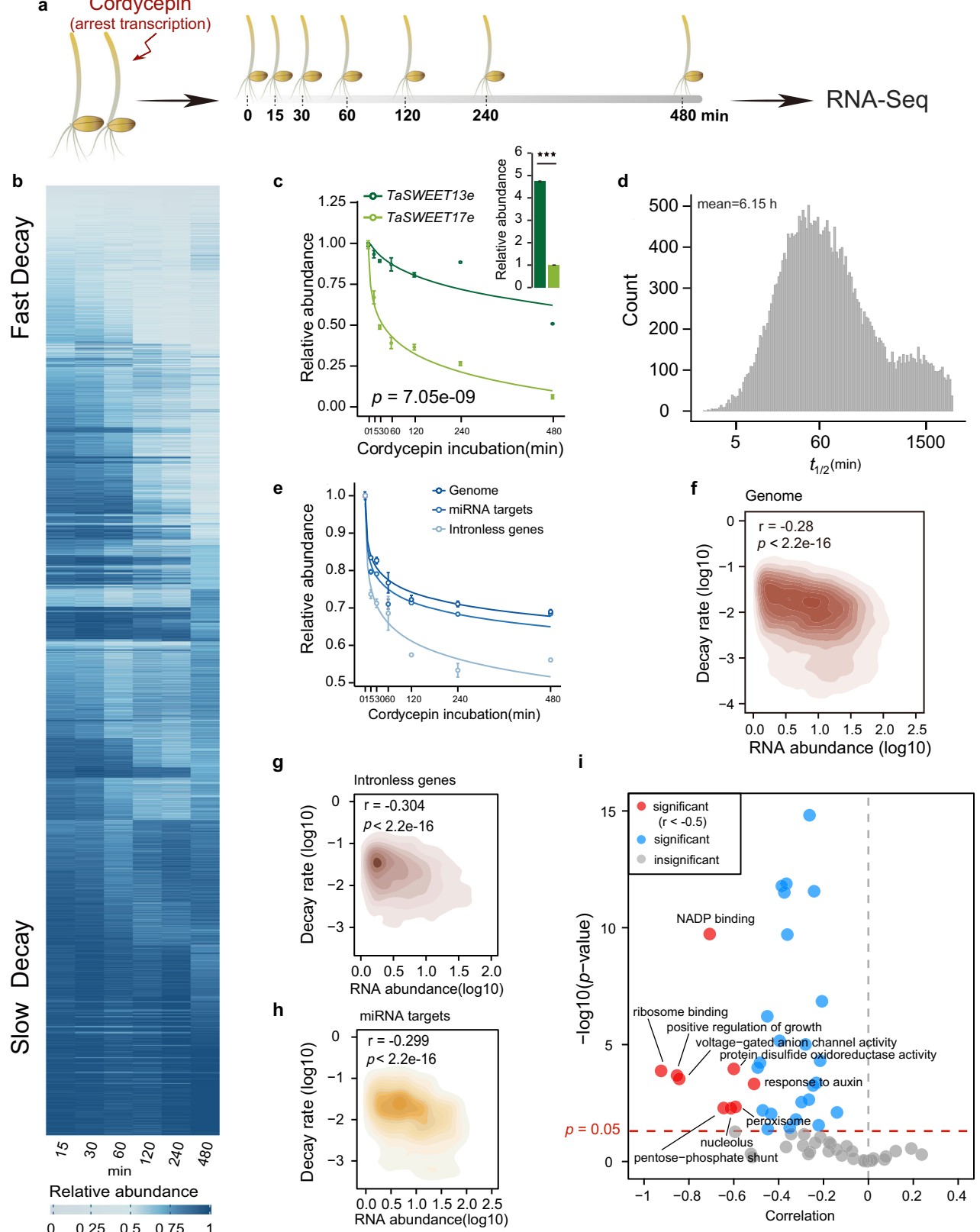

correlation coefficient was 0.33 ($p < 2.2e\text{-}16$), and for gene pairs with lower decay rates in A over B, the correlation coefficient was 0.27 ($p < 2.2e\text{-}16$) (Fig. 2d, e). Our gene ontology (GO) analysis revealed that both A and B subgenomes with longer half-lives tend to be linked to mutual biological functions such as metabolism, protein synthesis, and energy (Fig. 2f). Notably, biological functions associated with cellular

development and maintenance, such as transcription, plant growth, and cell division are specifically enriched with A subgenome mRNAs with longer half-lives and slower decay rates (Fig. 2f, pink dots). In contrast, those mRNAs in the B subgenome linked to longer half-lives and slower decay rates are specifically involved in biological functions such as oxidoreductase activity, response to oxidative stress, and

**Fig. 1 | mRNA decay landscape in tetraploid Kronos. a** Schematic of determining mRNA stability in wheat by transcriptional arrest analysis with cordycepin treatment. **b** Hierarchical clustering of genes with different decay rates, based on the transcript abundance at the different time courses of transcription inhibition. The decay rate was determined using a mathematical modeling approach based on the maximum likelihood method applied to data obtained with the cordycepin inhibitor method[30]. The genes with a fast decay rate and low RNA abundance are shown in light blue, while the genes with a slow decay rate and high RNA abundance are shown in dark blue. The heatmap displays fold changes in RNA abundance (RPM) at each time point relative to 0 min. The relative decay rate at 0 min is set to 1. **c** *TaSWEET13e*, and *TaSWEET17e* from the same gene family with different decay rates are shown as an example. The bar plot represents the steady-state relative

mRNA abundance (error bars indicating ± SEM (standard error of the mean), $n_{replicate\ samples} = 3$; ***$p < 0.001$, $p = 4.7e-06$ by one-sided student's $t$-test), while the line plot shows the degradation trend of the gene pair (error bars indicating ± SEM, $n_{replicate\ samples} = 3$; $p = 7.05e-09$ by one-sided repeated measures ANOVA test). **d** The distribution of mRNA half-lives at the genome-wide level. The mean half-life in Kronos is 6.15 h. **e** RNA abundance curves for transcripts from the whole transcriptome, miRNA target, and intronless genes, respectively (error bars indicating ± SEM, $n_{replicate\ samples} = 3$). **f–h** The correlation between decay rates and RNA abundance (**f**: $r = -0.28$, $p < 2.2e-16$; **g** $r = -0.304$, $p < 2.2e-16$; **h** $r = -0.299$, $p < 2.2e-16$, two-sided Pearson correlation test). **i** The correlation between decay rates and RNA abundance for functional gene categories (significant: $p < 0.05$, by two-sided Pearson correlation test).

hydrogen peroxide catabolism (Fig. 2f, green dots). These functions are typically involved in cellular responses to stress and environmental changes. Our collective findings indicate that RNA stability displays subgenomic asymmetry, significantly contributing to their steady-state expression level asymmetry. The subgenomic asymmetry at the RNA stability level exhibits divergent preferences for gene pairs involved in different biological functions.

## Several mRNA features influence the stability of wheat mRNA

Many mRNA features such as transcript length, GC and AU content, codon usage, and mRNA folding features have been reported to affect mRNA stability in other species[13,15–18,20,35]. Therefore, we undertook a comprehensive analysis to determine whether mRNA features also affect mRNA stability in wheat. Firstly, using the two-sided Pearson correlation test we found a relatively weak positive correlation between decay rates and mRNA lengths, particularly the length of coding sequences (CDS) (Supplementary Fig. 5a, CDS, $r = 0.18$, $p < 2.2e-16$). In general, we found that the longer the mRNA, the more rapidly it degrades. Our findings are similar to previous studies in other species[13,25,31]. The GC content in the 5′ UTR showed a weak negative correlation with the decay rate, while the AU content in the 5′ UTR showed a positive correlation with the decay rate (Supplementary Fig. 5b (GC content) and 5c (AU content), 5′ UTR, $r = -0.19$ and 0.19, $p < 2.2e-16$, respectively), suggesting that sequence content in the 5′ UTR may be associated with RNA decay rates. Recent studies have shown that translation impacts mRNA stability in a codon-dependent manner[14,18], so we measured the relationship between translation efficiency (TE) and RNA decay rates[34]. Using the two-sided Pearson correlation test, we found a weak negative correlation where mRNAs with higher translation efficiency tend to be more stable (Supplementary Fig. 5d, $r = -0.17$, $p < 2.2e-16$). Subsequent gene classification based on GO analysis allowed for the investigation of the correlation between translation efficiency and RNA decay rate within specific functional categories. Notably, genes involved in proton transmembrane transport and mitochondrial functions displayed the most significant negative correlations, underscoring their tendency towards heightened translation and stability (Supplementary Fig. 5e and Supplementary Data 4). We then analyzed whether the codon adaptation index (cAI) and the tRNA adaptation index (tAI) influence RNA stability and found no significant correlation between cAIs and decay rates and a significant negative correlation between tAIs and decay rates (Supplementary Fig. 5f (cAI), g (tAI), $r = -0.01$ and $-0.05$, $p = 0.03$ and 1.3e-14, respectively). Next, we measured whether individual codon occurrences can influence RNA decay rates and used the codon stabilization coefficient (CSC) method[17]. With CSC, codons that have a positive correlation with mRNA decay rates are enriched in more stable mRNAs. Conversely, codons that have a negative correlation with mRNA decay rates are enriched in less stable mRNAs. Out of the 61 codons, we found 33 and 22 codons that showed significant positive and negative correlations, respectively, with mRNA decay rates (Supplementary Fig. 5h). Notably, CSC values ranged from −0.14 to 0.16, indicating that codons only have a weak influence on RNA stability.

Taken together, mRNA length, sequence content, and codon have subtle but significant influences on mRNA stability. We also systematically classified factors such as transcript and UTR length, sequence content, and intron number on variations in RNA decay rates among homoeologous genes. For instance, we found that there are 1955 homoeoglous gene pairs exhibiting a negative correlation between decay rate and transcript length, while 1841 pairs show a positive correlation. Additionally, 2003 pairs demonstrate a negative correlation between the decay rate and AU content of transcripts, whereas 1891 pairs show a positive correlation. Moreover, in 3169 homoeologous gene pairs, we observed a negative correlation between decay rate and tAI, and in 3143 pairs showing a positive correlation. Furthermore, in 1036 pairs, homoeologs with more introns compared to their counterparts had lower decay rates, while in 982 pairs, those with more introns exhibited faster decay rates (Supplementary Data 5).

## Systematic discovery of RNA structural motifs that govern mRNA stability in wheat

Previous studies in other species have reported that mRNA structures in the 3′ UTR tend to be associated with mRNA stability[23–29]. We utilized our wheat in vivo RNA structurome[34] to assess whether RNA structures also influence mRNA stability in wheat. We calculated SHAPE reactivities, which measure the single-strandedness of individual nucleotides, and then correlated them with our mRNA decay rates using the two-sided Pearson correlation test. We found that only the average SHAPE reactivities of 3′ UTRs from different genetic intervals showed a significant positive association with mRNA decay rates (Fig. 3a, $r = 0.27$, $p = 5.3e-09$). We then calculated the base-pairing probability (BPP) for each nucleotide corresponding to these in vivo RNA structures. We consistently observed a significant negative correlation between the average BPPs of the 3′ UTRs and our mRNA decay rates (Supplementary Fig. 6, $r = -0.21$, $p = 0.004$). In contrast to other mRNA features, the RNA structure in 3′ UTRs has a stronger influence on mRNA stability, indicating that it may serve as a major determinant.

Based on this observation, we then performed a systematic identification of RNA structural motifs associated with mRNA stability in wheat using the computational framework pyTEISER v1.0.0 (Pythonic Tool for Eliciting Informative Structural Elements in RNAs)[36]. pyTEISER utilizes in vivo RNA structure information and sequence information to identify cis-regulatory elements that govern various RNA-related processes, such as splicing, RNA processing, and steady-state gene expression[36]. We incorporated our wheat in vivo RNA structure experimental data into the pyTEISER framework[34,36]. We also incorporated the RNA structural motif generated by FOREST[37] to expand the range of RNA structural motif seeds. We discovered 118 strong RNA structural motifs in 3′ UTRs that passed our statistical threshold[23,36] for the most likely RNA structural motifs contributing to mRNA stability (Fig. 3b, c and Supplementary Data 6). Among these 118 stability-associated RNA structural motifs, 64 were termed sRSMs (stable RNA structural

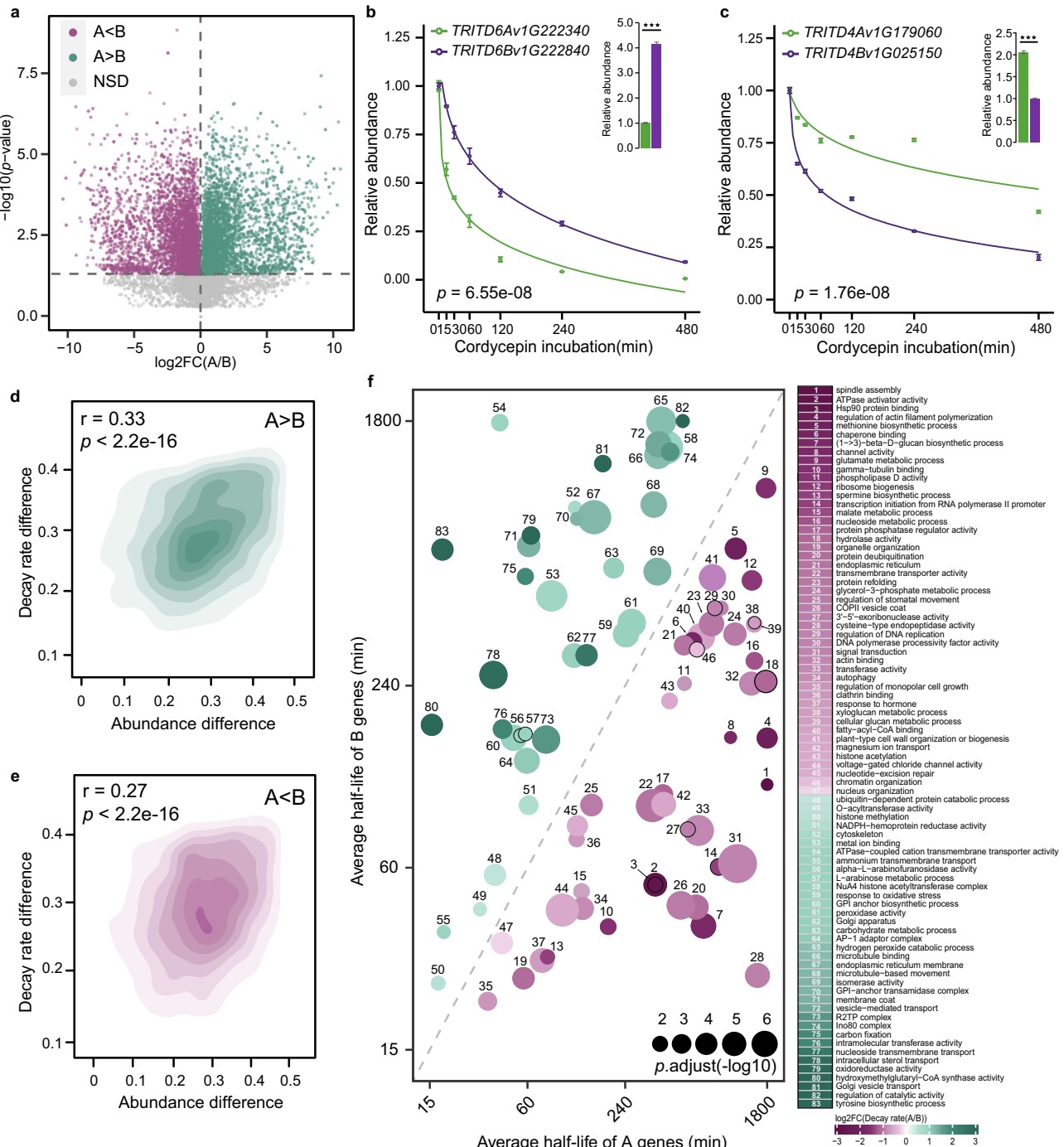

**Fig. 2 | mRNA decay subgenome asymmetry. a** Scatter plot illustrating the variation in decay rates between the A and B subgenomes in Kronos. The decay rates of A subgenome genes show either significantly higher or lower values compared to those in the B subgenome, or exhibit no significant difference between the two subgenomes. (A < B, $n_{pairs}$ = 3414; A > B, $n_{pairs}$ = 3184; NSD, no significant difference, $n_{pairs}$ = 6350, by one-sided student's $t$-test, FC, fold change). **b, c** Representative examples illustrating mRNA decay subgenome asymmetry. The bar plot represents the steady-state relative mRNA abundance (***$p$ < 0.001 by one-sided student's $t$-test). The line plot depicts the degradation trend of the gene pairs (error bars indicating ± SEM, $n_{replicate\ samples}$ = 3; **b** $p$ = 6.55e-08, **c** $p$ = 1.76e-08, one-sided repeated measures ANOVA test). **d, e** The correlations between the differential mRNA decay rates and the differential steady-state mRNA abundances for homoeologous gene pairs. ($r$ = 0.33 and 0.27, $p$ < 2.2e-16, two-sided Pearson correlation test). **f** Gene ontology (GO) analysis. The half-life of each point corresponds to the average of the half-lives of homoeologous gene sets in subgenome A and B within the corresponding enriched GO categories. The size of the points represents the degree of enrichment for the corresponding GO term. The colors in the GO functional heatmap are determined by the fold change values of the $A_{decay\ rate}$/$B_{decay\ rate}$.

motifs) due to their pattern with increasing mRNA stability; while 54 were classified as uRSMs (unstable RNA structural motifs) due to their pattern with decreasing mRNA stability (Fig. 3b, c and Supplementary Data 6). Next, we compared the structural features between sRSMs and uRSMs. Interestingly, sRSMs tend to be

significantly more stable than uRSMs (Supplementary Fig. 7a). sRSMs also contain significantly more canonical base pairs (A−U and G−C base pairs) than uRSMs (Supplementary Fig. 7b). Altogether, our findings reveal that RNA structure plays an important role in affecting mRNA stability in wheat.

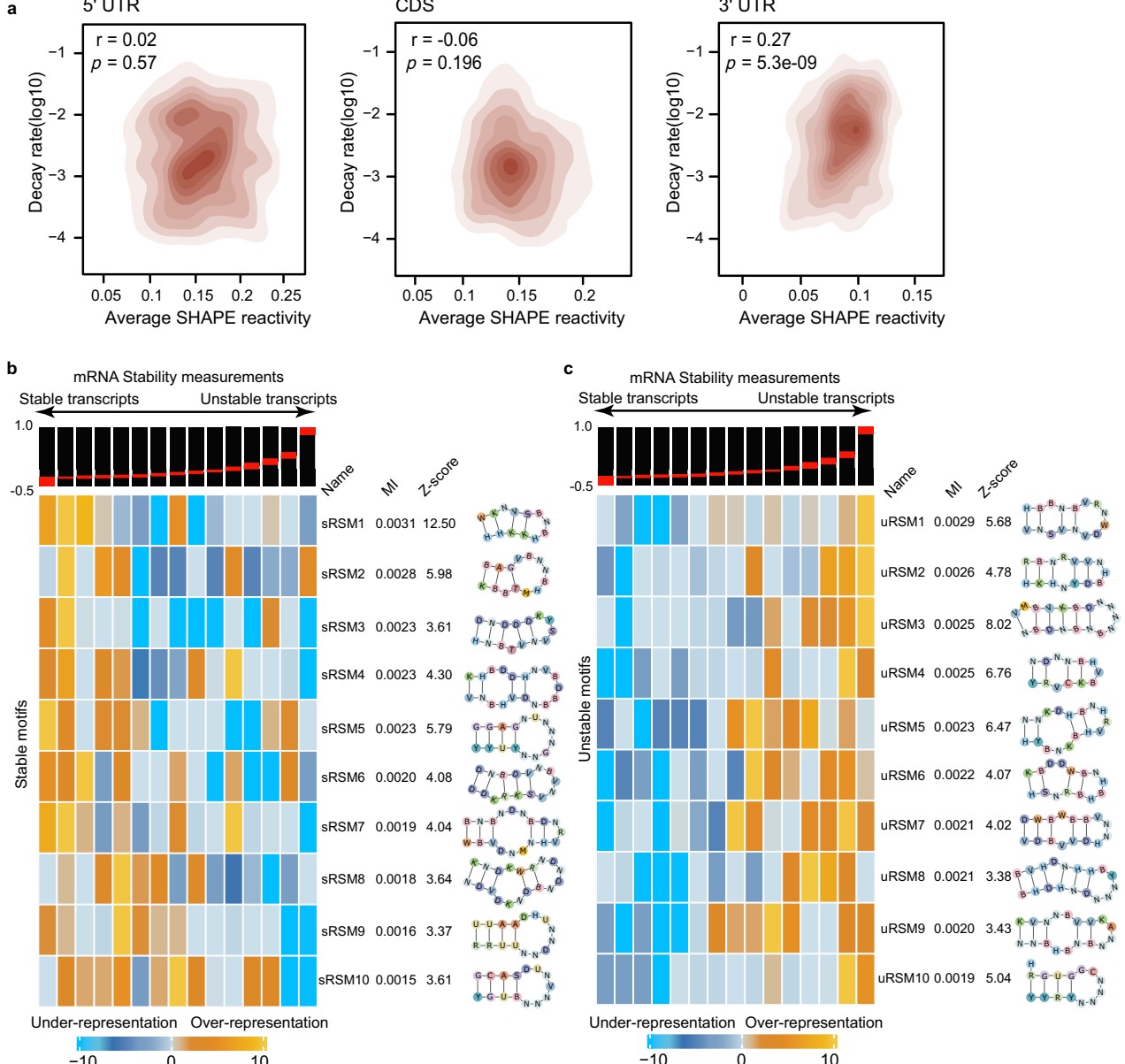

**Fig. 3 | Systematic discovery of RNA structural motifs that govern mRNA stability in wheat. a** The correlations between mRNA decay rate and the average SHAPE reactivity in different genetic regions (two-sided Pearson correlation test). **b** The example stable RNA structural motifs (sRSMs). The transcripts are divided equally into 15 bins from left (highly stable) to right (highly unstable) according to their stability measures (indicated by the red line). The heatmap demonstrates the enrichment of ten example sRSMs. The colors indicate the enrichments of individual sRSM in the individual bin. **c** The example unstable RNA structural motifs (uRSMs), otherwise in (**b**).

## The subgenomes of wheat exhibit distinct preferences for stability-associated RNA structural motifs

We then assessed whether subgenomes have preferences for different stability-associated RNA structural motifs. We found that our sRSMs and uRSMs exhibit varying degrees of enrichment in the A and B subgenomes, respectively (Fig. 4a and Supplementary Fig. 8a). We used the dual-luciferase reporter assay to validate those subgenomic sRSMs that appeared to be favored. In the homoeologous pair, the 3′ UTR of *TRITD1Av1G039750* from the A subgenome contains two stable RNA structural motifs, sRSM2 and sRSM32, which are absent in the 3′ UTR of the B subgenome *TRITD1Bv1G052450* gene (Fig. 4b). This A subgenome gene *TRITD1Av1G039750* is significantly more stable than the B subgenome gene, *TRITD1Bv1G052450* (Fig. 4b and Supplementary Fig. 9a). By fusing the 3′ UTRs of this homoeologous gene pair with the coding region of the Firefly reporter gene along with an identical Renilla luciferase as an internal control, we determined that the decay rate of *TRITD1Av1G039750*'s 3′ UTR, which possesses two sRSMs, is significantly slower than that of *TRITD1Bv1G052450*'s 3′ UTR, in which these sRSMs are absent (Fig. 4c, *p* = 2.53e-04, one-sided repeated measures ANOVA test). In another homoeologous gene pair, the 3′ UTR of the B subgenome gene *TRITD4Bv1G019060* contains one stable RNA structural motif, sRSM15, while the 3′ UTR of the A subgenome gene *TRITD4Av1G187460* lacks this sRSM (Fig. 4d; Supplementary Fig. 9b). Using the reporter assay experiment, we determined that the decay rate of *TRITD4Av1G187460*'s 3′ UTR is significantly faster than that of *TRITD4Bv1G019060*'s 3′ UTR (Fig. 4e, *p* = 3.48e-04, one-sided repeated measures ANOVA test). To further validate the function of these sRSMs, we performed the reporter assay using the structural motifs, and motifs with mutations

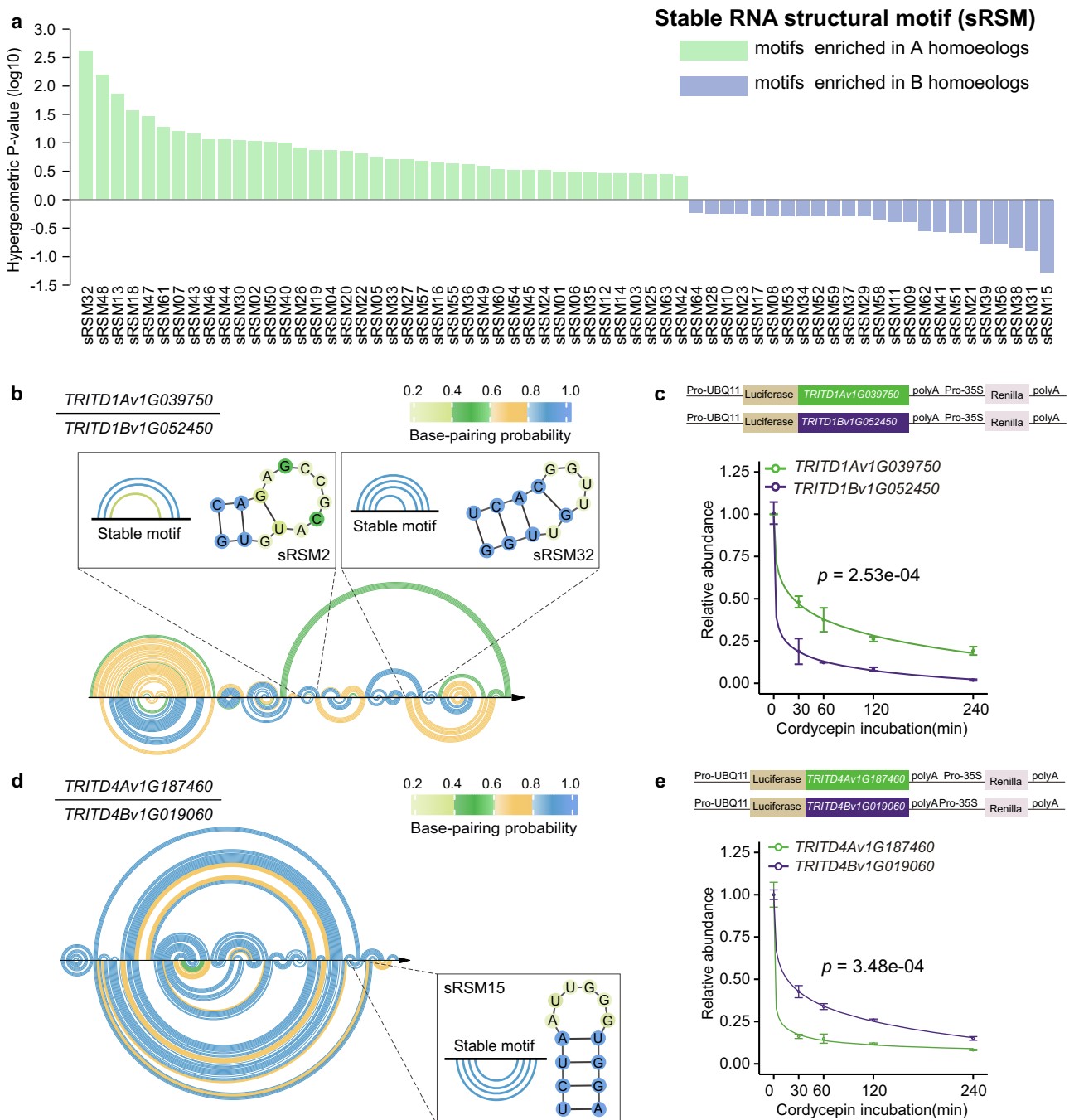

**Fig. 4 | The subgenomes of wheat exhibit distinct preferences for stability-associated RNA structural motifs. a** The different enrichments of the stable RNA structural motifs (sRSMs) in the A and B subgenomes. **b** An example of subgenomic sRSMs. In the homoeologous gene pair, the 3′ UTR of *TRITD1Av1G039750* from the A subgenome contains two stable RNA structural motifs, sRSM2 and sRSM32, which are absent in the 3′ UTR of the B subgenome *TRITD1Bv1G052450* gene. The arc plot shows the RNA structure in the 3′ UTR of the homoeologous pair, with each arc representing one base pair colored according to the base-pairing probability. The structures of sRSM2 and sRSM32 were illustrated separately as shown. **c** The experimental

validation of the mRNA stabilities for the homoeologous pair in (**b**) using the dual-luciferase reporter assay. (error bars indicating ± SEM, $n_{replicate\ samples}$ = 3, $p$ = 2.53e-04, one-sided repeated measures ANOVA test). **d** Another example where the 3′ UTR of the B subgenome gene *TRITD4Bv1G019060* contains one stable RNA structural motif, sRSM15, while the 3′ UTR of the A subgenome gene *TRITD4Av1G187460* lacks this sRSM, otherwise in (**b**). **e** The experimental validation of the mRNA stabilities for the homoeologous gene pair in (**d**) using the dual-luciferase reporter assay. (error bars indicating ± SEM, $n_{replicate\ samples}$ = 3, $p$ = 3.48e-04, one-sided repeated measures ANOVA test).

to disrupt or rescue the structures. We used the short fragments of the three structural motifs (sRSMs) and designed their corresponding disrupted and rescued mutated structural motifs (Supplementary Data 7). The decay rates of the disrupted sRSMs were significantly faster than those of their corresponding sRSMs (Supplementary Fig. 10, sRSM2 vs sRSM2_disrupted: $p$ = 3.52e-05; sRSM32 vs

sRSM32_disrupted: $p$ = 1.59e-03; sRSM15 vs sRSM15_disrupted: $p$ = 2.35e-04, one-sided repeated measures ANOVA test). Additionally, the decay rates of the rescued sRSMs were significantly slower than those of their corresponding disrupted sRSMs, but similar to those of their corresponding sRSMs (Supplementary Fig. 10, sRSM2 vs sRSM2_rescued: $p$ = 0.0825; sRSM32 vs sRSM32_rescued: $p$ = 0.201;

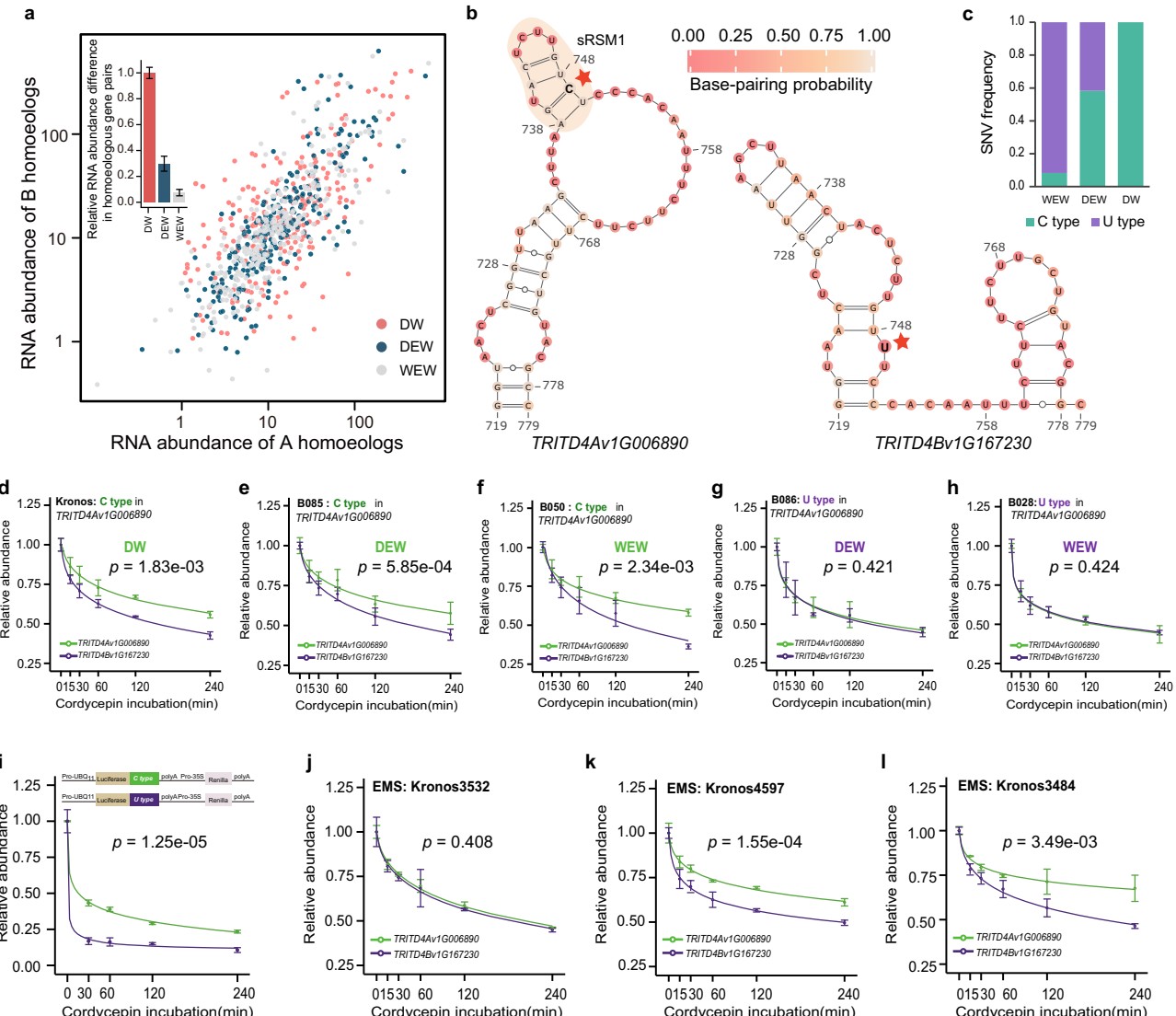

**Fig. 5 | Stability-associated RNA structural motifs may have evolved during wheat domestication and be altered in the EMS-induced mutants. a** The correlations of RNA abundances between A and B subgenomes of the homoeologous gene pairs containing the subgenomic preferred sRSMs in Kronos, durum wheat. The differential steady-state RNA abundances of those homoeologous gene pairs are dramatically reduced in both domesticated emmer and wild emmer wheat accessions (DEW: B086 and WEW: B028), where the SNVs disrupted the subgenomic preferred sRSMs. The bar plot represents the average differential expressions between subgenomes (error bars indicating ± SEM, $n_{replicate\ samples}$ = 3). **b** In the 3' UTR of *TRITD4Av1G006890* in Kronos, there is one SNV, C749 inside the sRSM1 that forms a CG base pair (marked with red asterisks). This C-type changes to U-type in the 3' UTR of *TRITD4Bv1G167230*, resulting in the disruption of the sRSM1. **c** The proportions of the C-type and U-type present in the 3' UTR of

*TRITD4Av1G006890* in the WEW, DEW, and DW assessions as reported[2]. **d**–**h** The corresponding RNA stability measured by qRT-PCR across accessions with different types of SNVs. The line plots depict the degradation trend of the gene pair (error bars indicating ± SEM, $n_{replicate\ samples}$ = 3, **d** $p$ = 1.83e-03, **e** $p$ = 5.85e-04, **f** $p$ = 2.34e-03, **g** $p$ = 0.421, **h** $p$ = 0.424, one-sided repeated measures ANOVA test). **i** This SNV alone was capable of altering mRNA stability. The RNA decay rates of the fused 3' UTR regions containing only this C-type or U-type SNV with the dual-luciferase reporter gene were measured by qRT-PCR (error bars indicating ± SEM, $n_{replicate\ samples}$ = 3, $p$ = 1.25e-05 by one-sided repeated measures ANOVA test). **j**–**l** The RNA decay rates of the gene pair were measured by qRT-PCR across three EMS-mutagenized mutants of Kronos (Kronos3532, Kronos4597, and Kronos3484, error bars indicating ± SEM, $n_{replicate\ samples}$ = 3, **j** $p$ = 0.408, **k** $p$ = 1.55e-04, **l** $p$ = 3.49e-03, one-sided repeated measures ANOVA test).

sRSM15 vs sRSM15_rescued: $p$ = 0.0965, one-sided repeated measures ANOVA test). These results further support the functional significance of the sRSMs on RNA stability.

We then asked whether the subgenomic preferences for stability-associated RNA structural motifs might be important for the subgenomic asymmetry of gene expression. Interestingly, we found much stronger correlations between the differential RNA decay rates and the differential steady-state RNA levels in the homoeologous gene pairs possessing stability-associated RNA structural motifs (Supplementary Fig. 8b, c, decay rate, A < B, $r$ = 0.38, $p$ = 7.5e-6; A > B, $r$ = 0.47, $p$ = 1.6e-6). These findings suggest that the subgenomes may adopt different RNA stability-associated structural motifs to regulate

their subgenomic RNA stability, resulting in subgenomic differences in steady-state mRNA abundances.

## Stability-associated RNA structural motifs may have evolved during wheat domestication

Large numbers of SNVs were selected during wheat domestication. Durum wheat (DW), *Triticum turgidum* L. ssp. *durum* is *a* type of cereal grain utilized for the production of pasta and couscous[38]. DW was developed from its domesticated form, emmer wheat (DEW), *Triticum turgidum* L. ssp. *dicoccum*. DEW was initially derived from its wild form, emmer wheat (WEW), *Triticum turgidum* L. ssp. *dicoccoides*. WEW was one of the first cereals to be domesticated in the Fertile Crescent. Our

previous work has shown that SNVs that alter RNA structures tend to be more strongly selected during wheat domestication[34]. As such, we hypothesized that SNVs located in the 3′ UTRs that alter stability-associated RNA structural motifs may have played a significant role during wheat domestication. As we could only directly assess the SNVs that perturb RNA structure, we focused on those SNVs that alter the subgenomic preferred sRSMs in 3′ UTRs. Subsequently, we calculated the fixation index ($F_{ST}$) to measure the species differentiation across three wheat populations[2] and found that the $F_{ST}$ values of those SNVs that alter subgenomic preferred sRSMs were significantly higher than other SNVs (Supplementary Fig. 11).

We then asked whether these selected SNVs may affect the subgenomic differences of the steady-state mRNA abundances by perturbing stability-associated structural motifs. We selected one representative WEW accession (B028) and one representative DEW accession (B086), accordingly. We performed RNA-seq experiments on these accessions and determined the steady-state expression levels of the corresponding homoeologous gene pairs containing the subgenomic preferred sRSMs in Kronos, a durum wheat (DW). We found that the differential steady-state RNA abundances of those homoeologous gene pairs are dramatically reduced in both domesticated emmer and wild emmer wheat accessions, where the SNVs disrupted the subgenomic preferred sRSMs (Fig. 5a and Supplementary Fig. 12). Therefore, our results suggest that during domestication wheat may have adopted SNVs to alter stability-associated RNA structural motifs to regulate the subgenomic differences of steady-state mRNA abundances.

We further demonstrate the adoption of stability-associated RNA structural motifs during domestication through the example of the homoeologous gene pair, *TRITD4Av1G006890* and *TRITD4Bv1G167230*. Between the two subgenome genes, there is one SNV, C749 inside the sRSM1 that forms a CG base-pair in the 3′ UTR of *TRITD4Av1G006890*. However, this C-type changes to U-type in the 3′ UTR of *TRITD4Bv1G167230*, resulting in the disruption of the sRSM1 (Fig. 5b). *TRITD4Av1G006890* is much more stable than *TRITD4Bv1G167230* in our wheat RNA decay data. Notably, this C-type in *TRITD4Av1G006890* was observed in all DW accessions and 58% of DEW accessions, while 92% of WEW accessions possessed the U-type nucleotide (Fig. 5c). We validated our RNA stability findings using qRT-PCR in the U-type DEW and WEW accessions. In Kronos and C-type DEW and WEW accessions, we confirmed our wheat RNA decay data which showed that the A subgenome had a slower decay rate than the B subgenome (Fig. 5d–f). In the U-type DEW and WEW accessions, we found that both the A and B subgenomes exhibited similar RNA decay rates (Fig. 5g, h). To validate that this SNV alone was capable of altering mRNA stability, we fused the 3′ UTR regions containing only this C-type or U-type SNV with the dual-luciferase reporter gene (Fig. 5i). We then performed our transcription arrest assay for measuring RNA decay rates and found that the decay rate of the C-type SNV was significantly slower than that of the U-type (Fig. 5i). This result further supports that this single SNV is capable of changing mRNA stability. To strengthen our findings, we examined an additional homoeologous gene pair, *TRITD4Av1G172030*, and *TRITD4Bv1G035170*, harboring an SNV, A1861, within sRSM12. This SNV forms an AU base-pair in the 3′ UTR of *TRITD4Av1G172030*, but changes to a G-type in the 3′ UTR of *TRITD4Bv1G035170*, disrupting the sRSM12 (Supplementary Fig. 13a). Notably, this A-type was found in all DW and DEW accessions, while 67% of WEW accessions had the G-type (Supplementary Fig. 13b). qRT-PCR validated our RNA stability results, showing that the A subgenome had a slower decay rate than the B subgenome in A-type Kronos, A-type DEW, and A-type WEW accessions (Supplementary Fig. 13c–e). In the G-type WEW accession, both subgenomes had similar RNA decay rates (Supplementary Fig. 13f). Altogether, our results indicate that this validated SNV is likely to have been selected during wheat domestication, leading to subgenomic differences in steady-state mRNA

abundances through the adoption of stability-associated structural motifs.

Over recent decades, an extensive number of point mutations induced by EMS have been deliberately introduced into wheat to enhance crop diversity[39]. The majority of these EMS-induced mutations are situated within non-coding regions. Subsequently, we tested whether EMS-induced mutations located in non-coding regions, such as the 3′ UTR, could potentially influence mRNA stability. Among our Kronos EMS mutant library, we successfully identified one EMS mutant that has a C to U mutation in the 3′ UTR of *TRITD4Av1G006890*, which disrupted another CG base-pair close to the sRSM1 (Fig. 5j and Supplementary Fig. 14). Notably, we found that in this EMS mutant, both the A and B subgenomes showed similar decay rates (Fig. 5j). In contrast, two additional EMS mutants carrying SNVs in the 3′ UTR of *TRITD4Av1G006890*, which do not alter RNA structures, maintained similar subgenomic differences in RNA decay rates between the A and B subgenomes (Fig. 5k, l and Supplementary Fig. 14). Our results indicate that RNA structure analysis provides a valuable perspective for studying the large number of EMS-induced mutations located in non-coding regions.

## Discussion

### Our wheat RNA decay landscape uncovers diverse mRNA stabilities and subgenomic asymmetry across the wheat transcriptome

For decades, gene regulation in wheat has mainly focused on transcriptional regulation. However, recent evidence on steady-state RNA abundance in wheat throughout developmental stages suggests the importance of mRNA stability[40]. Here, our study reveals a detailed landscape of wheat RNA decay, providing a comprehensive view of the stability of the wheat transcriptome and the basis for post-transcriptional gene regulation. Overall, the half-lives of wheat mRNAs, spanning over 33,430 transcripts, exhibit a diverse range from minutes to hours, with a mean value of 6.15 h. Notably, the mean half-life of wheat appears slightly longer compared to that observed in *Arabidopsis*[30,31]. The distribution of half-lives in wheat is wider than that in *Arabidopsis*[30,31], indicating that polyploid wheat cultivars have a broader range of decay rates. This may also be attributable to the increase in the number of mRNAs. Consistent with prior studies in other species[31,32], both intronless genes and miRNA-targeted mRNAs demonstrate lower stability compared to other types of mRNAs in wheat. Furthermore, intronless genes and miRNA target transcripts have shorter half-lives than the whole genome set, with over 70% of these transcripts having half-lives of less than 3 h, which is comparable to that found in *Arabidopsis* (Supplementary Fig. 4). Similarly, the half-life distribution of both intronless genes and miRNA targets are wider than those reported in *Arabidopsis*[31], further suggesting diverse decay rates across the wheat transcriptome (Supplementary Fig. 4). Based on the diversity of decay rates in wheat, it will be interesting for future studies to dissect further the decay rates for individual isoforms using long-read sequencing platforms.

Importantly, the correlation between mRNA decay rates and steady-state mRNA abundances is up to −0.28, underscoring the significant impact of mRNA stability on overall mRNA levels. Furthermore, a notable 6598 pairs of homoeologous genes (up to 13,196 mRNAs) display significantly distinct decay rates. Remarkably, the differential decay rates are also closely associated with the differences in steady-state mRNA abundances (Fig. 1f–i). In contrast to other factors contributing to the asymmetry of steady-state expression levels, such as histone modification and variation in transposable elements[10], mRNA stability exerts the most significant influence on subgenomic gene expression. This emphasizes the critical importance of post-transcriptional regulation in governing gene expression. Moreover, A subgenomic long-life mRNAs predominantly participate in basic functions such as transcription, plant growth, and cell division,

whereas B subgenomic long-life mRNAs are more involved in cellular stress responses (Fig. 2f and Supplementary Fig. 2). These subgenomic preferences suggest that the divergence of subgenomic mRNA stability may have evolved to uphold heterotic interactions and eliminate redundancies between subgenomes.

### Contrary to the subtle yet significant influences of other factors, RNA structure in the 3′ UTRs plays a major role in governing mRNA stability in wheat

Our comprehensive analysis of various mRNA features, including transcription length, sequence composition, codon usage, and RNA structures, offers a holistic understanding of their contributions to mRNA stability. In wheat, we observed that longer CDSs generally exhibit higher decay rates (Supplementary Fig. 5a). Long CDS regions tend to contain more pre-mature termination codons (PTCs), which induce nonsense-mediated decay[41]. Additionally, sequence composition in the 5′ UTR appears to strongly influence mRNA stability, with mRNAs containing higher GC content in the 5′ UTR demonstrating greater stability, while those with higher AU content in the 5′ UTR are less stable (Supplementary Fig. 5b, c). We did not observe significant sequence enrichment in the CDS and 3′ UTR associated with RNA decay rate (Supplementary Fig. 5b, c). A previous study in *Arabidopsis* did not find any significant correlation between AU/GC sequence content and mRNA decay rates in all genic regions[31]. Notably, AU sequence features enriched in the 5′ UTR are consistent with a previous report in *Arabidopsis* on the sequence features enriched in uncapped mRNAs[42]. Possibly, sequence motifs do not act alone. Furthermore, we also found that wheat mRNAs with high translation efficiencies tend to display greater stability, likely due to their preference for specific codons (Supplementary Fig. 5d). It is also possible that ribosome protects mRNAs from decay enzymes[43]. However, the relatively weak correlation indicates that the interrelationships between the two processes are more complex. Our further analysis of the codon occurrences showed that 55 out of 61 codons significantly but weakly associate with mRNA decay rates (Supplementary Fig. 5h). Moreover, analysis of our wheat in vivo RNA structurome dataset revealed a significant positive correlation between decay rates and SHAPE reactivities only in 3′ UTRs, indicating that mRNAs with more single-stranded regions in the 3′ UTR are generally less stable (Fig. 3a). This is consistent with previous observations in *Arabidopsis*, rice, human and yeast[25,27,28,44,45]. This is likely because the majority of endonucleases are single-stranded binding proteins, where more single-stranded regions tend to be targeted by endonucleases[46–48]. In contrast, another type of in vivo RNA structure profiling method using chemical probing with a reverse transcription mutation assay in both *Arabidopsis* and rice showed that RNA structures in 3′ UTRs, measured by Gini index, significantly anti-correlate with mRNA transcription levels ($r = -0.12$)[29]. The study suggested that certain RNA structures in 3′ UTRs destabilize mRNAs via earlier polyadenylation[29]. These contrasting results may be due to the different methods used to measure RNA structures[49]. It is also possible that 3′ UTR contains diverse RNA structure features that are associated with different regulatory processes, such as exonuclease activities, endonuclease activities, and RNA processing[50–52].

### Systematic identification of stability-associated RNA structural motifs reveals subgenomic preferences

We systematically identified 118 stability-associated RNA structural motifs using pyTEISER v1.0.0, a statistical algorithm that integrates both genome-wide RNA structure profiling data and RNA decay data, to determine RNA structural motifs in 3′ UTRs that are significantly informative of genome-wide mRNA stability (Fig. 3b, c and Supplementary Data 6). Notably, of these 118, 64 were stable while 54 were unstable motifs (Fig. 3b, c and Supplementary Data 6). By assessing the base-pairing properties and folding strength, these stable motifs tend to have more canonical base pairs and lower free energy per nucleotide than unstable motifs (Supplementary Fig. 7a, b). In general, we found that RNA molecules tend to adopt intricately folded structures rather than remaining single-stranded, primarily to mitigate the risk of non-specific endonuclease cleavage. Thus, RNA structural motifs are distributed throughout the entire RNA molecule. Here, we comprehensively assessed the genome-wide RNA structurome in wheat and determined the RNA structural motifs that confer mRNA stability, thus providing direct regulatory elements for modulating mRNA stability. Moreover, we identified distinct preferences of stable and unstable RNA structural motifs between the two subgenomes (Fig. 4a and Supplementary Fig. 8a). This suggests potential evolutionary divergence of stability-associated RNA structural motifs, despite limited sequence variation, adding an additional layer to subgenomic divergence.

### The subgenomic asymmetry of mRNA stability in wheat can be shaped by both natural and induced mutations

The process of crop domestication has been profoundly influenced by single nucleotide variations (SNVs)[6,9]. SNVs have served as genetic markers that distinguish between wild and domesticated varieties. As most SNVs occur in non-coding regions, more attention has been increasingly given to their potential effect on RNA structures, impacting post-transcriptional regulation of gene expression[53]. The evolution of tetraploid wheat driven by human activity encompasses domestication (WEW to DEW) and ongoing evolution during domestication (DEW to DW)[38]. A large number of SNVs were identified during this domestication process[2]. Notably, we postulate that those SNVs that perturbed our identified stable RNA structural motifs may have been progressively selected during domestication (Supplementary Fig. 11). Our further assessment of a wild emmer wheat accession (B028) and a representative domesticated emmer wheat accession (B086) revealed a significant reduction in global subgenomic differences in gene expression in the absence of these structure-perturbing SNVs (Fig. 5a). Individual sRSM validation further supported that a single nucleotide selected during domestication, which altered RNA structure resulted in a change in mRNA stability (Fig. 5b–h). Our results highlight how evolutionarily selected SNVs can fine-tune stability-associated RNA structural motifs, thus contributing to subgenomic gene expression diversification during domestication. With the increasing utilization of EMS-induced mutations in wheat to enhance diversity, we further expanded our demonstration by assessing the available Kronos EMS population[39]. One EMS mutant contains the nucleotide variation that perturbed the CG base pair close to sRSM1 altering the stability of the subgenomic mRNA (Fig. 5j and Supplementary Fig. 14). The other two EMS mutants that contain the nucleic variations that did not affect RNA structural base-pairs remained similar subgenomic asymmetry of mRNA stabilities in Kronos (Fig. 5k, l and Supplementary Fig. 14). As both natural and EMS-induced mutants are predominantly located in non-coding regions, our study provides a valuable perspective for assessing the impact of these non-coding mutations, particularly in 3′ UTRs, on post-transcriptional regulation of gene expression. A thorough examination of in vivo RNA structuromes across diverse wheat populations will facilitate the discovery of how stability-associated RNA structure motifs are impacted by SNVs and the corresponding effects on RNA stability, particularly in genes associated with agronomic traits, to inform breeding strategies. In summary, our study reveals the promising potential of RNA structure-mediated gene regulation as an effective avenue for crop improvement efforts.

## Methods

### Plant materials and growth conditions

The seeds of *Triticum turgidum* L. ssp. *durum* (cv. Kronos) and EMS-mutagenized mutants of Kronos (Kronos3532, Kronos4597, and Kronos3484) were obtained from the John Innes Centre Germplasm

Resource Unit. The Kronos EMS library was published by the Dubcovsky Lab in 2017[39]. Researchers are able to access this database online (dubcovskylab.ucdavis.edu/wheat_blast and www.wheat-tilling.com), where they can search for specific mutations in their target genes. Plants were crossed to obtain homozygotic mutant alleles. The other accessions of tetraploid wheat *Triticum turgidum* L. ssp. *dicoccum* (cv. B089, cv. B086, cv. B085) and *dicoccoides* (cv. B050, cv. B028) were kindly provided by Dr. Fei Lu. All seeds were germinated in water for 4 days at 22 °C in the dark before treatment.

### Transcriptional arrest

Transcriptional arrest analysis was performed as described with modifications[30]. Wheat seedlings were placed on a double layer of filter paper in a petri dish and incubated in a buffer containing cordycepin (1 mM PIPES, pH 6.25, 15 mM sucrose, 1 mM potassium chloride, 1 mM sodium citrate, and 1 mM cordycepin). For the reporter assay in tobacco (*N. benthamiana*) leaves, the seedlings were harvested and cut into small disks (~5 mm diameter) after 48 h of agroinfiltration, followed by treatment with cordycepin buffer. Tissue samples were collected at various time intervals (15 min, 30 min, 60 min, 120 min, 240 min, 480 min). Subsequently, the samples were promptly frozen in liquid nitrogen and stored at −80 °C until RNA extraction. Plants that did not undergo cordycepin incubation were harvested as controls, representing 0 min of cordycepin exposure. RNA extraction was carried out, and the samples were subjected to either quantitative real-time PCR (qRT-PCR) or RNA sequencing.

### RNA-seq data mapping and analysis

The seedlings of wheat (Kronos, B089, B086, B085, B050, B028), both treated with time-course transcriptional arrests and untreated, were directly applied to RNA extraction using the Qiagen RNeasy Plant Mini Kit. The extracted RNAs were used for strand-specific library generation by BGI Genomics, following the manufacturer's BGISEQ-500 protocol. For RNA-seq experiments (Kronos, B086, B028), three independent biological replicates were assessed. We performed the polyA selection and conducted the 150 bp pair-end read sequencing on the BGISEQ-500 platform with strand-specific RNA sequencing. Hisat2 v2.1.0[54] was used for mapping the reads to the durum wheat genome assembly (Svevo RefSeq 1.0)[38] corrected by accession-specific SNVs information from previous study[2]. Based on the uniquely mapped read counts, we calculated per million mapped reads (RPMs) for measuring the abundance of each gene in each sequencing library.

### Decay profile normalization and modeling of mRNA decay

Both the normalization of transcript read abundances and the modeling of mRNA decay rates were reported by Sorenson et al.[30]. In order to correct the abundance of the total pool of RNA, the selected decay factors are elaborated in Supplementary Data 8. The genes selected as the decay factor have the following characteristics: (1) highly expressed; (2) stable[30]. Normalized decay profiles based on $T_{0min}$ and decay factor were utilized for modeling decay rates. The *RNAdecay* v1.16.0 (R package from Bioconductor) was used to model mRNA decay rates[30]. During the modeling process, the mod_optimization function in the *RNADecay* package was used to obtain statistical results. The model with the minimum value of AICc (Akaike information criterion) was considered the optimal model for each gene. For every decay rate, we calculated the half-life using the formula: $t_{1/2} = \ln(2)/\text{rate}$[30].

### Analysis for intronless genes and miRNA targets

We calculated the number of introns for the genes of tetraploid Kronos based on the .gff3 file. We defined the genes with a count of zero introns as "intronless genes" for downstream analyses. Then, we employed TargetFinder v1.7[55], a computational prediction tool, to predict miRNA target sites on the transcripts in Kronos. The 126 mature miRNA sequences from wheat in FASTA format were obtained from the miRBase database (http://miRbase.org/). The intronless genes and miRNA targets are listed in Supplementary Data 3.

### Quantitative real-time PCR

The reverse transcription process utilized the *TransScript* One-Step gDNA Removal and cDNA Synthesis SuperMix to synthesize cDNA from total RNA employing an oligo dT primer. Subsequent real-time PCR analysis was carried out using the Applied Biosystems QuantStudio 5 quantitative real-time PCR Systems with THUNDERBIRD™ SYBR® qPCR Mix. Two housekeeping genes *RLI* (in wheat) and *Actin* (in tobacco), were used to normalize the expression data. Detailed primer information is provided in Supplementary Data 7.

### Gene ontology analysis

GO annotation terms for *Triticum turgidum* L. ssp. *durum* genes were obtained from the Ensembl Plants Genes 55 database (https://plants.ensembl.org/biomart/martview/). GO enrichment analysis was performed using *clusterProfiler 4.0*[56], and GO analysis was conducted with the assistance of the AmiGO[57] to investigate the relationship between mRNA half-life and gene functions.

### Base pairing probability (BPP) calculation

Base pairing probability (BPP) in RNA secondary structure prediction is derived from the partition function, which sums the weighted contributions of all possible configurations using a dynamic programming algorithm[58]. In this manuscript, the BPPs are calculated using the RNAfold from the ViennaRNA package with the SHAPE constraint and the inclusion of the "-p" parameter, representing the equilibrium binding probabilities of base pairs across possible RNA secondary structures[59].

### Systematically identify stability-associated RNA structural motifs

To systematically identify stability-associated RNA structural motifs, we adopted pyTEISER v1.0.0, a computational framework for identifying cis-regulatory elements that govern various RNA-related processes[36]. The initial step in identifying RNA structural motifs using pyTEISER involves defining RNA structural motif seeds, which are short stem-loop elements meeting the following criteria: stem length between 4 and 7 nucleotides, loop length between 4 and 9 nucleotides, information content between 14 bits and 20 bits, and number of degenerate nucleotides in the seed between 4 and 6. In addition to the RNA structural motifs derived from the context-free grammars in pyTEISER v1.0.0, we also incorporated the FOREST approach to generate an additional seed library of RNA structural motifs based on thermodynamic folding algorithm[37], expanding the range of RNA structural motif seeds. These RNA structural motifs were filtered by our in vivo RNA structure experimental data[36].

Subsequently, 3' UTR RNAs with valid RNA decay data were divided equally into 15 groups according to their decay rates from low to high. The RNA structural motifs were then calculated separately for their mutual information ($MI(R;A)$), where $R$ represents the presence of an RNA structural motif and $A$ represents its absence, for each decay group. The $P$ value of MI and the *Z-score* were calculated based on 1000 randomization-based statistical tests, following the instructions of pyTEISER[36].

$$MI(R;A) = \sum_{r \in R} \sum_{a \in A} p(r,a) \log\left(\frac{p(r,a)}{p(r)p(a)}\right)$$

$$Z-score = \frac{MI(R,A) - \frac{1}{n}\sum\left(MI_{shuffled}(R,A)\right)}{\sigma_{MI_{shuffled}(R,A)}}$$

Then, those RNA structural motifs with an MI greater than 0.001, and *Z-score* greater than 2 were selected as stability-associated RNA structural motifs. Among them, if the frequency of the RNA structural motif negatively correlated with the degradation rate, then it was regarded as an RNA structural motif that accelerates RNA degradation (uRSMs). Conversely, if the frequency was positively correlated, it was regarded as an RNA structural motif that stabilizes RNAs (sRSMs).

**Plasmid construction and dual-luciferase reporter assay**
The 3′ UTRs of three pairs of homoeologous genes (*TRIT-D1Av1G039750* vs *TRITD1Bv1G052450*, *TRITD4Av1G187460* vs *TRITD4Bv1G019060*, and *TRITD4Av1G006890* vs *TRITD4Bv1G167230*) were amplified from wheat cDNAs using CloneAmp HiFi PCR Premix (Clontech). We also used the short fragments of the three structural motifs (sRSM2, sRSM32, and sRSM15) and designed their corresponding disrupted and rescued mutated structural motifs. After amplification, the PCR products were incorporated into the expression vector inter2, which had been cleaved with BamHI and SmaI, using In-Fusion (Clontech) technology. Sequencing-validated vectors were then introduced into *Agrobacterium tumefaciens* GV3101 and infiltrated into tobacco leaves (*N. benthamiana*). The gene expression levels were measured using a dual-luciferase reporter assay, following the established protocol[34]. Detailed sequences of the primers and constructs can be found in Supplementary Data 7.

**Statistics and reproducibility**
No statistical method was used to predetermine the sample size. No data were excluded from the analyses. The experiments were not randomized. The investigators were not blinded to allocation during experiments and outcome assessment.

**Reporting summary**
Further information on research design is available in the Nature Portfolio Reporting Summary linked to this article.

## Data availability
The transcriptional arrest libraries from this study can be downloaded from the NCBI Sequence Read Archive under SRA accession SRP409681 of BioProject PRJNA901641. Meanwhile, RNA-seq libraries of B028, B086, and Kronos accessions are available at NCBI Genebank (SRA) under the following accession numbers: SRP527122 (BioProject PRJNA1148332) for B086 and B028, and SRP315863 (BioProject PRJNA72321934)[34] for the libraries of Kronos_RNAseq_replicate. Source data are provided with this paper.

## Code availability
The custom scripts and workflows used in our manuscript are available from https://github.com/Huakun-Lab/RNA_Decay_in_wheat.

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

## Acknowledgements

The study is supported by the National Key Research and Development Program of China (2022YFF1003303 to B.L.), the National Key Research and Development Program of China (2021YFF1000900 to H.Z.), the National Natural Science Foundation of China (32170229 and 32000178 to H.Z.), the Fundamental Research Funds for the Central Universities (2412023YQ005 to H.Z.), the United Kingdom Biotechnology and Biological Sciences Research Council (BBSRC) (BB/X01102X/1 to B.Y., H.Y., Y.Z., and Y.D.); a BBSRC DTP studentship (2578674 to B.Y.); the European Research Council (ERC) (selected by the ERC, funded by BBSRC Horizon Europe Guarantee EP/Y009886/1 to Y.D.); a Human Frontier Science Program Fellowship (LT001077/2021-L to H.Y). This research was supported in part by the NBI Computing Infrastructure for Science (CiS) group and the Informatics platform. We thank the John Innes Centre Germplasm Resources Unit for supplying wheat seeds. We also thank Dr. Simon Griffiths, Prof. Graham Moore, and Prof. Cristobal Uauy for their suggestions.

## Author contributions

H.Z. and Y.D. conceived the study. H.W., H.Y., Y.Z., B.Y., B.L., Y.D., and H.Z. designed the study. H.W., Y.Z., W.S., Q.L., L.R., and Y.L. performed the experiments. H.Y., B.Y., and H.W. performed the analyses. H.Z. and Y.D. supervised the analyses. H.W., H.Y., B.Y., B.L., Y.D., and H.Z. wrote the manuscript with input from all authors.

## Competing interests

The authors declare no competing interests.
