## [Peer Review File · Nature Communications]

REVIEWER COMMENTS

Reviewer #1 (Remarks to the Author):

This study demonstrates that RNA structure, particularly in the 3' UTR region, significantly affects RNA stability. Identifying specific RNA structural motifs can provide deeper insights into the mechanisms influencing RNA stability. This study identifies subgenome-specific RNA structural motifs (RSMs) and their association with RNA stability, providing a valuable example and introducing new insights from plant research into the RNA structure field. Additionally, the discovery that SNVs in sRSM regions are a significant driver of population differentiation is a notable highlight. I have several suggestions to further improve the manuscript.

Major comments:

1. The results in Sections 1 and 2 are overly descriptive and lack exciting highlights. It would be better if they were as concise, logical, and content-rich as Section 5. For example, Fig 1g should also plot the correlation of decay rates according to different gene types (Genome, miRNA targets, Intronless genes). Additionally, based on the number of each gene type shown in Fig 1f, many other types of genes (n=26,323) should be categorized and analyzed in detail, potentially revealing stronger correlations.
2. Figures 4b-e are nice experiments. However, the 3'UTR is too long to confirm the function of the short RNA structural motif. The authors should perform the assay using the structural elements, and elements with mutations to break or rescue the structure, to validate the function of the RNA structural motif.
3. Regarding Figure 5: First, the whole transcriptome analysis is limited. More meta-analyses should be performed to confirm the conclusion; one perfect example is not enough. Second, for Figures 5e-f, what about the C-type DEW and WEW between A and B (TRITD4Av1G006890/TRITD4Bv1G167230)? Third, in Lines 271-274, what are the subgenomic differences in steady-state mRNA abundances among DW, DEW, and WEW? Are SNVs enriched in the subgenomic differences RNA, particularly the SNVs that perturb RNA structures?

Minor comments:

1. In line 74, the full name of "EMS" should be provided since this is the first occurrence of the word in this manuscript.
2. L91: "heat map" should be "heatmap". L342: "humans" should be "human". There are repetitive sentences in Fig S5.

3. L124, please add more description or definition about the A subgenome and B subgenome.
4. L443, L463: Add software version information.
5. In lines 153 to 155, the author used 'significant' and 'notably' to describe weak correlations ($r=-0.19$ and $r=0.19$), which may be inappropriate. Also in line 159, "significant" was used to describe a weak correlation ($r=-0.17$).
6. In line 180, methods about base-pairing probability (BPP) calculation should be cited, and the details of the method for calculating BPP from in vivo RNA structures should be explained in the METHOD section.
7. Fig4b-c, the SHAPE score and SHAPE score differences between A and B homoeologs should be shown to highlight the structural contribution in RNA Structural Motifs (RSM).
8. The significance of Fig 4c,4e, and also 5d,5g,5i-j.

Reviewer #2 (Remarks to the Author):

In this manuscript, Wu et al. examined mRNA stability in a cultivar of tetraploid durham wheat using the transcription inhibitor cordycepin at a series of time points. They estimated mRNA decay rates and compared them between homoeologous genes in each of the subgenomes to reveal about 6500 homeologous gene pairs with higher decay rates from one subgenome. They also found significant positive correlations between differential mRNA decay rates and differential steady state mRNA abundances of homoeologous gene pairs between the A and B subgenomes. This is the first study of mRNA stability in the context of a polyploid plant, to my knowledge. It provides new perspectives on what causes unequal expression of homoeologous genes in polyploids. I think this study will inspire interest among researchers in doing similar research with other polyploid plants such as canola and cotton.

I have never done mRNA stability assays, so I was not able to critically evaluate their methods. Hopefully another reviewer did that.

I suggest starting the abstract with a sentence about mRNA stability instead of the sentence about advances in deep sequencing, because the manuscript is about that conceptual topic. Also, the Introduction starts by talking about genomics techniques. I recommend that it start with the conceptual topic of mRNA stability.

Line 382: “and a represented domesticated emmer wheat accession”. Do they mean a representative accession?

Reviewer #3 (Remarks to the Author):

Wu and co-authors presented a novel insight into wheat RNA decay and the impact of RNA structure on the regulation of RNA decay. This study provides a new understanding of post-transcriptional regulation in polyploid crops. The differential decay rate of subgenomic regions presents another level of gene regulation during wheat domestication. Overall, it is a very interesting study that will provide new resources and knowledge to the plant science community.

I have several comments as follows:

1. The authors presented that miRNA targets have a faster decay rate. Since miRNA-target interactions depend on sequence complementarity, is there any divergence in sequence complementarity between the A and B subgenome that leads to subgenomic differences in RNA decay?
2. I am curious to know how the authors performed mRNA decay analysis because different transcript isoforms may have different decay rates for one single gene. Were the decay rates of all the transcripts merged together? We can claim bona fide mRNA decay only if analyses were performed at the isoform level. The authors may discuss the possibility of decay rate analysis at the transcript isoform level in the Discussion.
3. The authors discovered an interesting correlation between translation efficiency and RNA decay. It will be interesting that the authors could perform cluster these correlations and identify which genes have both high translation efficiency and high decay rates.
4. The authors found that intronless genes tend to degrade faster. I wonder if intronless genes contain more destabilized RNA structure motifs.
5. The authors identified homoeologous gene pairs with differential decay rates and also showed that mRNA length, sequence content and codon have impact on mRNA stability. I am wondering how many homoeologous genes are possibly affected by each of these mentioned factors. Kindly please provide these details. In addition, have the authors taken into consideration that many homoeologous genes have different exon-intron structures or different lengths of UTRs?

6. The authors defined the enrichments of stabilized and destabilized motifs across the transcriptomes. I wonder whether the number of these motifs matters more in RNA decay or if one single motif could have a strong impact.

7. The authors found that some variants subjecting to domestication selection contribute to subgenomic asymmetry of mRNA stability. I am interested to know whether some of these variants are associated with a specific agronomic trait. It will be very interesting to see trait variation between different plants results from genomic non-coding variants that lead to mRNA stability. Possible examples may be given or discussed in the Discussion section.

8. Some discussions on the balance between translation efficiency and RNA decay are important. The authors should add these in the manuscript.

9. The authors claimed the potential application of using RNA structure-mediated RNA decay regulation in breeding. The authors should discuss how this could be practically fulfilled in crop breeding in their discussion session.

Reviewer #4 (Remarks to the Author):

In this paper, Wu et al. conducted a time-course RNA sequencing (RNAseq) analysis over seven time points in durum wheat (*Triticum turgidum* L. ssp. durum, BBA) following cordycepin treatment to study transcriptional arrest. Their findings suggest a subgenomic asymmetry in mRNA stability. By integrating their results with previously published RNA structure profiling dataset, the authors explored the impact of 3' UTR structure and RNA structural motifs on mRNA stability. Additionally, they discovered that single-nucleotide variations (SNVs) influencing RNA structural motifs and subgenomic mRNA stability were selected during domestication, subsequently altering the steady-state expression levels. Overall, this study offers valuable insights into the role of structure-mediated RNA half-life in regulating RNA abundance.

However, given that this study heavily relies on integrated omics analysis, a major concern is the incomplete description of the bioinformatic methods and data availability. Additionally, the data presentation is also unclear, which needs significant improvement. It is also recommended that the customized code for data reproducibility be made available on GitHub.

Bellowed is the detail for the major concerns:

Figure S1a demonstrated the good reproducibility of the three replicates for each time point using Pearson correlation coefficients. Could the authors visualize these 21 samples using a PCA plot? It would be interesting to examine the overall similarity among these samples.

Figure S1b shows the read numbers for each library. However, it is unclear whether these are total reads, mapped reads, or uniquely mapped reads. Additionally, the mapping rate should be provided.

3. Lines 420-429: The description of the method for RNA-seq data mapping and analysis is very ambiguous. For the strand-specific library construction, it should be clarified whether they are polyA-selected. For the sequencing parameters, it is important to mention whether the reads are paired-end or single-end and what the read length is. These details should be included in the methods section instead of the results. For the data analysis, it is essential to specify which tool and parameters were used for read mapping. Additionally, how the authors dealt with reads mapping to both homeologous gene pairs needs to be explained. The number of subgenome-specific reads used to measure the differential abundance between homeologous gene pairs should also be provided. And totally how many pairs of homeologous genes were detected in the 0 time point? This section is crucial for understanding how subgenomic asymmetry of homeologous gene expression is measured.

4. In the legend for Figure 1b, the heatmap shows labels for fast decay and slow decay on the left. Is the heatmap ordered by mRNA decay rate? How is the decay rate defined? Neither the figure legend nor the Decay Profile Normalization method describes this detail clearly. The color bar and figure legend indicate that the values in the heatmap represent the relative transcript abundance over six time points, but do not include the 0-minute point. Are these values the transcript abundance (RPM) or the fold change in RNA abundance at each given time point relative to the 0-minute point? If so, this should be clearly described in the figure legend.

5. Lines 145-185: The authors investigated the correlation between each mRNA feature and the stability of wheat mRNA individually. A supervised learning approach could be used to deduce the feature importance by using all the features as input for modeling the mRNA decay rate.

6. Why are intronless genes (intron number = 0) less stable? Does the number of introns affect RNA half-life?

7. Lines 246-255: Two accessions (how many replicates?) were selected for the RNA-seq experiment (Fig. 5a). However, these RNA-seq libraries were not described in the methods section, nor are they mentioned in the data availability statement.

8. Lines 275-286: One EMS mutant with a C to U mutation in the 3' UTR of TRITD4Av1G006890 was detected in a Kronos EMS library. However, no references related to this EMS library were provided. If this library has not been previously published, details about the Kronos EMS library should be described. Importantly, the method for detecting the mutant with the C to U mutation is also missing.

REVIEWER COMMENTS

Reviewer #1 (Remarks to the Author):

This study demonstrates that RNA structure, particularly in the 3' UTR region, significantly affects RNA stability. Identifying specific RNA structural motifs can provide deeper insights into the mechanisms influencing RNA stability. This study identifies subgenome-specific RNA structural motifs (RSMs) and their association with RNA stability, providing a valuable example and introducing new insights from plant research into the RNA structure field. Additionally, the discovery that SNVs in sRSM regions are a significant driver of population differentiation is a notable highlight. I have several suggestions to further improve the manuscript.

Response:

We are very grateful for the reviewer's constructive comments and valuable suggestions on our manuscript. We have fully addressed the comments and made significant improvements to our revised manuscript.

Major comments:

1. The results in Sections 1 and 2 are overly descriptive and lack exciting highlights. It would be better if they were as concise, logical, and content-rich as Section 5. For example, Fig 1g should also plot the correlation of decay rates according to different gene types (Genome, miRNA targets, Intronless genes). Additionally, based on the number of each gene type shown in Fig 1f, many other types of genes ($n=26,323$) should be categorized and analyzed in detail, potentially revealing stronger correlations.

Response:

We thank the Reviewer for the comments. We have modified Sections 1 and 2 based on the comments in our revised manuscript. We have also added the correlations of decay rates with RNA abundance for miRNA targets and intronless genes (**Rebuttal Fig. 1a, b and revised Fig. 1g, h**). Additionally, we have included a detailed classification of other types of genes according to their functions (**Rebuttal Fig. 1c and revised Fig. 1i**). We have reorganized the figures and moved some of them into the supplementary figures.

In the **Rebuttal Fig. 1a and b**, we plotted the correlations between decay rates and RNA abundances where we found that both intronless genes and miRNA targets also exhibit significant correlations ($r = -0.304$, $p < 2.2e-16$ and $r = -0.299$, $p < 2.2e-16$). Furthermore, we performed GO analysis and plotted the correlations for individual gene groups with each gene function (**Rebuttal Fig. 1c**). As shown, much stronger negative correlations were observed in the gene groups with functions associated with positive regulation of growth, response to auxin, voltage-gated anion channel activity, protein disulfide oxidoreductase activity, pentose-phosphate shunt, ribosome binding, and NADP binding (**Rebuttal Fig. 1c, and revised Supplementary Fig. 2**). These results indicate that RNA stability significantly contributes to the steady-state RNA abundances in these genes.

Rebuttal Fig. 1 (revised Fig. 1g, h, i) | Correlation analysis between decay rates and RNA abundance for different gene groups and clusters. a-c, Scatter plots illustrating the correlation between decay rate and RNA abundance among intronless genes (a), miRNA targets (b), and individual gene groups with each gene function (c) respectively, using the two-sided Pearson correlation test.

2. Figures 4b-e are nice experiments. However, the 3' UTR is too long to confirm the function of the short RNA structural motif. The authors should perform the assay using the structural elements, and elements with mutations to break or rescue the structure, to validate the function of the RNA structural motif.

Response:

We thank the Reviewer for the insightful comments. We have performed additional comprehensive experiments in validating the function of our stability-associated RNA structure motifs. We performed the reporter assay using the structural motifs, and motifs with mutations to disrupt or rescue the structures. We used the short fragments of the three structural motifs (sRSMs) and designed their corresponding disrupted and rescued mutated structural motifs (**Supplementary Data 7**). We then determined the corresponding decay rates. The results are consistent with our conclusion where we found that the decay rates of the disrupted sRSMs were significantly faster than those of their corresponding sRSMs (sRSM2 vs sRSM2_disrupted: $p = 3.52e-05$; sRSM32 vs sRSM32_disrupted: $p = 1.59e-03$; sRSM15 vs sRSM15_disrupted: $p = 2.35e-04$, one-sided repeated measures ANOVA test) (**Rebuttal Fig. 2a-c and revised Supplementary Fig. 10**). We also observed that the decay rates of the rescued sRSMs were significantly slower than those of their corresponding disrupted sRSMs, but similar to those of their corresponding sRSMs (sRSM2 vs sRSM2_rescued: $p = 0.0825$; sRSM32 vs sRSM32_rescued: $p = 0.201$; sRSM15 vs sRSM15_rescued: $p = 0.0965$, one-sided repeated measures ANOVA test) (**Rebuttal Fig. 2a-c and revised Supplementary Fig. 10**). Therefore, these results on the short fragments of the three structural motifs (sRSMs) are consistent with the results of the long 3' UTRs, further supporting the functional significance of the sRSMs on RNA stability.

Rebuttal Fig. 2 (revised Supplementary Fig. 10) | Decay rates of constructs with original, rescued, or disrupted motif structures across the sRSMs. The dark green line represents the decay trend for sRSM constructs with original motif structures, while the light green line represents the decay trend for sRSM constructs with motif structures rescued by redesigned sequence. The brown line indicates the decay trend for constructs with disrupted motif structures across all motifs. **a**, sRSM2 (sRSM2 vs sRSM2_disrupted: $p = 3.52e-05$, sRSM2 vs sRSM2_rescued: $p = 0.0825$, one-sided repeated measures ANOVA test); **b**, sRSM32 (sRSM32 vs sRSM32_disrupted: $p = 1.59e-03$, sRSM32 vs sRSM32_rescued: $p = 0.201$, one-sided repeated measures ANOVA test); **c**, sRSM15 (sRSM15 vs sRSM15_disrupted: $p = 2.35e-04$, sRSM15 vs sRSM15_rescued: $p = 0.0965$, one-sided repeated measures ANOVA test).

3. Regarding Figure 5: First, the whole transcriptome analysis is limited. More meta-analyses should be performed to confirm the conclusion; one perfect example is not enough. Second, for Figures 5e-f, what about the C-type DEW and WEW between A and B (TRITD4Av1G006890/TRITD4Bv1G167230)? Third, in Lines 271-274, what are the subgenomic differences in steady-state mRNA abundances among DW, DEW, and WEW? Are SNVs enriched in the subgenomic differences RNA, particularly the SNVs that perturb RNA structures?

Response:

We thank the Reviewer for the insightful comments.

In our previous analysis, we identified 304,835 SNVs between A and B subgenomes across the transcriptome¹. Based on our *in vivo* RNA structure profiling data in wheat seedlings, we have confidently identified 703 SNVs located in the 3'UTRs between A and B subgenomes that alter stability-associated RNA structural motifs. These SNVs tend to have significantly higher selections than other SNVs (**Supplementary Fig.11**). To further support our conclusion, we have conducted the experimental validation on another example (**Rebuttal Fig. 3, revised Supplementary Fig. 13**).

In the Rebuttal Fig.3, the homoeologous gene pair, *TRITD4Av1G172030* and *TRITD4Bv1G035170* contains a SNV, A1861, within sRSM12. This SNV forms an AU base-pair in the 3' UTR of *TRITD4Av1G172030*, but changes to a G-type in the 3' UTR of *TRITD4Bv1G035170*, disrupting of the sRSM12 (**Rebuttal Fig. 3a, Supplementary Fig. 13a**). Notably, this A-type in *TRITD4Av1G172030* was observed in all DW and DEW accessions, while 67% of WEW accessions possessed the G-type nucleotide (**Rebuttal Fig. 3b, Supplementary Fig. 13b**). To validate our RNA stability findings, we performed qRT-PCR in the A-type DW, A-type DEW, A-type WEW and G-type WEW accessions. In A-type DW, A-type DEW and A-type WEW accessions, we confirmed our wheat RNA decay data, which showed that the A subgenome had a slower decay rate than the B subgenome (**Rebuttal Fig. 3c-**

e, Supplementary Fig. 13c-e). In the G-type WEW accession, we found that both the A and B subgenomes exhibited similar RNA decay rates (**Rebuttal Fig. 3f, Supplementary Fig. 13f**).

Regarding our Fig. 5e-f, we have added the measurements of RNA decay rates in both C-type DEW and WEW accessions (**Rebuttal Fig. 4a, b; revised Fig. 5e, f**). These results strongly support our conclusion that the decay rate of the C-type A homoeologous gene is slower compared to that of the U-type B homoeologous gene in both C-type DEW and WEW accessions.

We further performed meta-gene analysis to assess the subgenomic differences in steady-state mRNA abundances among DW, DEW, and WEW accessions (B028: WEW, B086: DEW, and Kronos: DW, **Rebuttal Fig. 5, revised Supplementary Fig. 12**). We determined differentially expressed homoeologous gene pairs 1,120 in B028 (**Rebuttal Fig. 5a**), 1,709 in B086 (**Rebuttal Fig. 5b**), and 1,208 in Kronos (**Rebuttal Fig. 5c**) ($p < 0.05$, by one-sided Student's t -test), with 375 gene pairs common in all three accessions (**Rebuttal Fig. 5d**). We found that 43.66 % differentially expressed homoeologous gene pairs in B028 contains SNVs between A and B subgenomes; 47.22 % differentially expressed homoeologous gene pairs in B086 contains SNVs between A and B subgenomes; 52.40% differentially expressed homoeologous gene pairs in Kronos contains SNVs between A and B subgenomes. Based on our RNA structure data in Kronos, 21.03% differentially expressed homoeologous gene pairs in Kronos contains SNVs inside our identified stability-associated structural motifs.

Rebuttal Fig. 3 (revised Supplementary Fig. 13) | The stability-associated RNA structural motif in *TRITD4Av1G172030* may have evolved during wheat domestication. **a**, Diagram of the SNV, A1861, within sRSM12 in the 3' UTR of *TRITD4Av1G172030* (marked with red asterisks). This A-type nucleotide is replaced by a G in the 3' UTR of *TRITD4Bv1G035170*, disrupting the sRSM12 structure. **b**, The distribution of A- and G-type nucleotides in *TRITD4Av1G172030* across different

wheat accessions as reported². **c-f**, qRT-PCR validation showing the decay trends of A- and G-type mRNAs of *TRITD4Av1G172030* across DW, DEW, and WEW accessions (error bars indicating SE, $n_{\text{replicates}} = 3$, **c**: $p = 2.93\text{e-}05$, **d**: $p = 6.1\text{e-}07$, **e**: $p = 6.95\text{e-}06$, **f**: $p = 0.407$, one-sided repeated measures ANOVA test).

Rebuttal Fig. 4 (revised Fig. 5e, f) | Stability-associated RNA structural motifs may have evolved during wheat domestication and be altered in the EMS-induced mutants. **a**, The corresponding RNA stability measured by qRT-PCR across accessions with different types of SNVs. The line plot depicts the degradation trend of the gene pair (error bars indicating SE, $n_{\text{replicates}} = 3$, **a**: $p = 5.85\text{e-}04$, **b**: $p = 2.34\text{e-}03$, one-sided repeated measures ANOVA test).

Rebuttal Fig. 5 (revised Supplementary Fig. 12) | Analysis of subgenomic differences in steady-state mRNA abundances among wheat varieties. **a-c**, Scatter plots showing differentially expressed homoeologous gene pairs between A and B subgenomes in B028 (WEW) (**a**), B086 (DEW) (**b**), and Kronos (DW) (**c**) ($p < 0.05$, one-sided Student's t -test). **d**, Venn diagram illustrating the overlap of 375 common differentially expressed gene pairs across all three varieties.

Minor comments:

1. In line 74, the full name of "EMS" should be provided since this is the first occurrence of the word in this manuscript.

Response:

We thank the Reviewer for the comment. We have added the full name of EMS, ethyl methanesulfonate into our revised manuscript in line 76.

2. L91: "heat map" should be "heatmap". L342: "humans" should be "human". There are repetitive sentences in Fig S5.

Response:

We thank the Reviewer for the comment. We have made all the changes in our revised manuscript.

3. L124, please add more description or definition about the A subgenome and B subgenome.

Response:

We appreciate the reviewer's comment and have included additional details regarding the subgenome at its initial mention in the manuscript (Lines 40-43), as “In allopolyploids, a subgenome refers to one of the distinct sets of chromosomes that originates from one of the contributing species. Allopolyploidization process often induces significant genomic changes due to the merging of divergent genomes, leading to both structural and functional incompatibilities.”

4. L443, L463: Add software version information.

Response:

We thank the Reviewer for the comment. We have added the software versions in our revised manuscript: TargetFinder v1.7 (line 496) and pyTEISER v1.0.0 (line 523).

5. In lines 153 to 155, the author used 'significant' and 'notably' to describe weak correlations ($r=-0.19$ and $r=0.19$), which may be inappropriate. Also in line 159, “significant” was used to describe a weak correlation ($r=-0.17$).

Response:

We thank the Reviewer for the comment. We have revised our manuscript and changed the words to “weak” correlations.

6. In line 180, methods about base-pairing probability (BPP) calculation should be cited, and the details of the method for calculating BPP from in vivo RNA structures should be explained in the METHOD section.

Response:

We thank the Reviewer for the insightful comment. We have added details on the methods used to calculate BPP from *in vivo* RNA structures in the METHOD section of our revised manuscript.

“Base pairing probability (BPP) in RNA secondary structure prediction is derived from the partition function, which sums the weighted contributions of all possible configurations using a dynamic programming algorithm³. In this manuscript, the BPPs are calculated using the RNAfold from the ViennaRNA package with the SHAPE constraint and the inclusion of the “-p” parameter, representing the equilibrium binding probabilities of base pairs across possible RNA secondary structures⁴.”

7. Fig4b-c, the SHAPE score and SHAPE score differences between A and B homoeologs should be shown to highlight the structural contribution in RNA Structural Motifs (RSM).

Response:

We thank the Reviewer for the comment. We have added the SHAPE reactivities and differences between A and B subgenomes for Figure 4, as the **Rebuttal Fig. 6 (revised Supplementary Fig. 9)**.

Rebuttal Fig. 6 (revised Supplementary Fig. 9) | SHAPE reactivity profiles (top) and SHAPE reactivity differences (bottom) for the 3' UTRs of homoeologous gene pairs. a, *TRITD1Av1G039750* and *TRITD1Bv1G052450*. b, *TRITD4Av1G187460* and *TRITD4Bv1G019060*. Stable RNA structural motifs are highlighted in orange. The A and B subgenomes are aligned based on sequence alignment. The positions in the plot correspond to the positions of the A subgenome.

8. The significance of Fig 4c, 4e, and also 5d, 5g, 5i-j.

Response:

We thank the Reviewer for the comment. We have added the ANOVA test for measuring the significance for all the figures.

Reviewer #2 (Remarks to the Author):

In this manuscript, Wu et al. examined mRNA stability in a cultivar of tetraploid durum wheat using the transcription inhibitor cordycepin at a series of time points. They estimated mRNA decay rates and compared them between homoeologous genes in each of the subgenomes to reveal about 6500 homeologous gene pairs with higher decay rates from one subgenome. They also found significant positive correlations between differential mRNA decay rates and differential steady state mRNA abundances of homoeologous gene pairs between the A and B subgenomes. This is the first study of mRNA stability in the context of a polyploid plant, to my knowledge. It provides new perspectives on what causes unequal expression of homoeologous genes in polyploids. I think this study will inspire interest among researchers in doing similar research with other polyploid plants such as canola and cotton.

I have never done mRNA stability assays, so I was not able to critically evaluate their methods. Hopefully another reviewer did that.

Response:

We are very grateful for the reviewer's recognition on the importance of our work.

I suggest starting the abstract with a sentence about mRNA stability instead of the sentence about advances in deep sequencing, because the manuscript is about that conceptual topic. Also, the Introduction starts by talking about genomics techniques. I recommend that it start with the conceptual topic of mRNA stability.

Response:

We thank the Reviewer for the comment. We have revised our manuscript and emphasize more on the mRNA stability highlighted in the beginning of the abstract and introduction.

Line 382: “and a represented domesticated emmer wheat accession”. Do they mean a representative accession?

Response:

We thank the Reviewer for the comment. We have changed the words into “a representative domesticated emmer wheat accession”.

Reviewer #3 (Remarks to the Author):

Wu and co-authors presented a novel insight into wheat RNA decay and the impact of RNA structure on the regulation of RNA decay. This study provides a new understanding of post-transcriptional regulation in polyploid crops. The differential decay rate of subgenomic regions presents another level of gene regulation during wheat domestication. Overall, it is a very interesting study that will provide new resources and knowledge to the plant science community.

Response:

We are very grateful for the reviewer’s positive comments on our manuscript. We appreciate that the reviewer recognized our work as “*a very interesting study that will provide new resources and knowledge to the plant science community*”. We have fully addressed all the comments.

I have several comments as follows:

1. The authors presented that miRNA targets have a faster decay rate. Since miRNA-target interactions depend on sequence complementarity, is there any divergence in sequence complementarity between the A and B subgenome that leads to subgenomic differences in RNA decay?

Response:

We thank the Reviewer for the insightful comment. Indeed, we have identified six homoeologous gene pairs that show divergences in sequence complementarities between the A and B subgenomes, leading to subgenomic differences in RNA decay rates (**Rebuttal Fig 7**). Due to the limited number of homoeologous gene pairs, we focus on other factors in affecting subgenomic differences in this work.

Rebuttal Fig. 7 | Examples of different miRNA-target complementarities between two subgenomes. a-c, A subgenome genes' decay rates are more slowly than B subgenome genes. d-f, B subgenome genes' decay rates are more slowly than A subgenome genes. The sequences in the middle line represent the miRNA sequence information. The above sequences show the complementarity between the A subgenome genes and the miRNAs. The bottom sequences show the complementarity between the B subgenome genes and the miRNAs. The black nucleotide bases indicate identical base information between the A and B subgenomes, while red bases indicate SNVs between the A and B subgenomes, leading to different complementarities with the miRNAs.

2. I am curious to know how the authors performed mRNA decay analysis because different transcript isoforms may have different decay rates for one single gene. Were the decay rates of all the transcripts merged together? We can claim bona fide mRNA decay only if analyses were performed at the isoform level. The authors may discuss the possibility of decay rate analysis at the transcript isoform level in the Discussion.

Response:

We thank the Reviewer for the insightful comment. In our mRNA decay analysis, we used the longest isoform to calculate the mRNA decay rates for each gene. Due to the large size of wheat transcriptomes, we used the Illumina short-read sequencing platform to study RNA stability across transcriptomes. Thus, it is not possible to accurately distinguish between different isoforms. With the future development of PacBio and Nanopore in sequencing depth^{5, 6, 7}, it will be possible to conduct mRNA decay analysis using these long-read sequencing platform in the future. We added the future perspective in our discussion (lines349-351).

3. The authors discovered an interesting correlation between translation efficiency and RNA decay. It will be interesting that the authors could perform cluster these correlations and identify which genes have both high translation efficiency and high decay rates.

Response:

We thank the Reviewer for the insightful comment. We have clustered the genes into different groups with individual functions according to the gene ontology (GO) analysis. We then calculated the correlations between translation efficiency and RNA decay rate for individual gene groups (**Rebuttal Fig. 8, revised Supplementary Fig. 5e**). Our result revealed that gene functions involved in proton transmembrane transport and mitochondrial functions showed strong negative correlations, indicating that these genes tend to be highly translated and greatly stable (**Rebuttal Fig. 8, revised**

Supplementary Fig. 5e). We have also added a new supplementary table that provides a list of genes with both high translation efficiency (top 30%) and high decay rates (top 30%) (**Rebuttal Table 1, revised Supplementary Data 4**).

Rebuttal Fig. 8 (revised Supplementary Fig. 5e) | Correlation analysis between translation efficiency and RNA decay rate across gene clusters (significant, $p < 0.05$, two-sided Pearson correlation test).

4. The authors found that intronless genes tend to degrade faster. I wonder if intronless genes contain more destabilized RNA structure motifs.

Response:

We thank the Reviewer for the insightful comment. We performed Fisher’s exact test to compare the enrichments of both stable and unstable RNA structural motifs in intronless genes and genes with introns. We found no significant difference (odds ratio = 0.995, $p = 1$, **Rebuttal Fig. 9**) between the two groups. Both intronless genes and genes with introns contain similar proportions of stable and unstable motifs.

Rebuttal Fig. 9 | Comparison of sRSMs and uRSMs in intronless genes versus genome-wide genes. The Fisher's exact test was used to evaluate the differences in the proportions of sRSMs and uRSMs between intronless genes and genome-wide genes.

5. The authors identified homoeologous gene pairs with differential decay rates and also showed that mRNA length, sequence content and codon have impact on mRNA stability. I am wondering how many homoeologous genes are possibly affected by each of these mentioned factors. Kindly please provide these details. In addition, have the authors taken into consideration that many homoeologous genes have different exon-intron structures or different lengths of UTRs?

Response:

We thank the Reviewer for the insightful comment. In our analysis, we systematically investigated the differences in different factors that may affect the mRNA stability between homoeologous gene pairs, including transcript length, UTR lengths, sequence content, and intron numbers (**Rebuttal Table 2, revised Supplementary Data 5**). For instance, we found that there are 1,955 homoeologous gene pairs exhibiting a negative correlation between decay rate and transcript length, while 1,841 pairs show a positive correlation. Additionally, 2,003 pairs demonstrate a negative correlation between decay rate and AU content of transcripts, whereas 1,891 pairs show a positive correlation. Moreover, in 3,169 homoeologous gene pairs, we observed a negative correlation between decay rate and tAI, and in 3,143 pairs showing a positive correlation. Furthermore, in 1,036 pairs, homoeologs with more introns compared to their counterparts had lower decay rates, while in 982 pairs, those with more introns exhibited faster decay rates. In summary, these factors are all important for mRNA stability, but their effects vary greatly among different gene pairs. We added these information into our revised manuscript (lines 184-193).

6. The authors defined the enrichments of stabilized and destabilized motifs across the transcriptomes. I wonder whether the number of these motifs matters more in RNA decay or if one single motif could have a strong impact.

Response:

We thank the Reviewer for the insightful comment. We have performed an analysis to assess the number of stability-associated structure motifs that affect the RNA decay. We found that a trend of

decreasing decay rates as the number of sRSMs increases (**Rebuttal Fig. 10a**, 1 sRSM vs 2 sRSM: $p = 0.058$; 1 sRSM vs 3 sRSM: $p = 0.008$; 1 sRSM vs >3 sRSM: $p = 0.036$). Notably, there is a significant difference in decay rates between 3' UTRs with one and 3' UTRs with more than three sRSMs. The difference in decay rate between 3' UTRs with one and two sRSMs is not significant. Regarding uRSMs, we observed a significant increase in decay rates between 3' UTRs with one and two uRSMs, but no significant differences with 3' UTRs with three or more uRSMs (**Rebuttal Fig. 10b**, uRSM vs 2 uRSM: $p = 0.0005$; 1 uRSM vs 3 uRSM: $p = 0.064$; 1 uRSM vs >3 uRSM: $p = 0.844$). These results suggest that the number of stability-associated RNA structure motifs is likely important for mRNA stability.

Rebuttal Fig. 10 | Effect of RNA structural motifs' numbers on mRNA decay rates. a, The bar plot shows the mean decay rates of mRNAs with 3' UTRs containing varying numbers of sRSMs (one, two, three, and more than three sRSMs). **b**, The bar plot illustrates the mean decay rates of mRNAs with 3' UTRs containing varying numbers of uRSMs (one, two, three, and more than three uRSMs). The error bars indicate SE. Significance levels are: *** $p < 0.001$, ** $p < 0.01$, * $p < 0.05$, n.s. for no significance, by one-sided Student's t -test.

7. The authors found that some variants subjecting to domestication selection contribute to subgenomic asymmetry of mRNA stability. I am interested to know whether some of these variants are associated with a specific agronomic trait. It will be very interesting to see trait variation between different plants results from genomic non-coding variants that lead to mRNA stability. Possible examples may be given or discussed in the Discussion section.

Response:

We thank the Reviewer for the comment. In our work, as a proof of concept, we used the four-day seedlings wheat for our first investigation of RNA stability. It is unlikely to be associated with specific agronomic traits. We will study RNA stability in diverse developmental stages of wheat in our future studies in exploring the RNA stability-associated trait variation.

8. Some discussions on the balance between translation efficiency and RNA decay are important. The authors should add these in the manuscript.

Response:

We thank the Reviewer for the insightful comment. We have added the discussion on the balance between translation efficiency and RNA decay in our revised manuscript (lines 380-383).

9. The authors claimed the potential application of using RNA structure-mediated RNA decay regulation in breeding. The authors should discuss how this could be practically fulfilled in crop breeding in their discussion session.

Response:

We thank the Reviewer for the insightful comment. We have added the discussion on the potential crop breeding in our revised manuscript (lines 443-446).

Reviewer #4 (Remarks to the Author):

*In this paper, Wu et al. conducted a time-course RNA sequencing (RNAseq) analysis over seven time points in durum wheat (*Triticum turgidum* L. ssp. durum, BBAA) following cordycepin treatment to study transcriptional arrest. Their findings suggest a subgenomic asymmetry in mRNA stability. By integrating their results with previously published RNA structure profiling dataset, the authors explored the impact of 3' UTR structure and RNA structural motifs on mRNA stability. Additionally, they discovered that single-nucleotide variations (SNVs) influencing RNA structural motifs and subgenomic mRNA stability were selected during domestication, subsequently altering the steady-state expression levels. Overall, this study offers valuable insights into the role of structure-mediated RNA half-life in regulating RNA abundance. However, given that this study heavily relies on integrated omics analysis, a major concern is the incomplete description of the bioinformatic methods and data availability. Additionally, the data presentation is also unclear, which needs significant improvement. It is also recommended that the customized code for data reproducibility be made available on GitHub.*

Response:

We are very grateful for the reviewer's constructive comments and valuable suggestions on our manuscript. We have fully addressed the comments and revised our manuscript according to the reviewer's advice. We have provided detailed descriptions of both bioinformatic methods and data availability. We have also clarified and improved our data presentation. We have provided our customized code for data reproducibility, which is now available on GitHub (https://github.com/Huakun-Lab/RNA_Decay_in_wheat).

Bellowed is the detail for the major concerns:

1. Figure S1a demonstrated the good reproducibility of the three replicates for each time point using Pearson correlation coefficients. Could the authors visualize these 21 samples using a PCA plot? It would be interesting to examine the overall similarity among these samples.

Response:

We thank the Reviewer for the insightful comment. We have generated a PCA plot to visualize our 21 samples. Consistent with our correlation analysis, the PCA plot showed that the three biological replicates for each time point clustered closely together, indicating minimal variation among these biological replicates (**Rebuttal Fig. 11, revised Supplementary Fig. 1b**).

Rebuttal Fig. 11 (revised Supplementary Fig. 1b) | The PCA plot illustrates the clustering of RNA abundances in the three biological replicates for each time point.

2. Figure S1b shows the read numbers for each library. However, it is unclear whether these are total reads, mapped reads, or uniquely mapped reads. Additionally, the mapping rate should be provided.

Response:

We thank the Reviewer for the important comment. We have provided the detailed information: total reads count, total mapping rates and unique mapping rates for each library (**Rebuttal Table 3, revised Supplementary Data 1**).

3. Lines 420-429: The description of the method for RNA-seq data mapping and analysis is very ambiguous. For the strand-specific library construction, it should be clarified whether they are polyA-selected. For the sequencing parameters, it is important to mention whether the reads are paired-end or single-end and what the read length is. These details should be included in the methods section instead of the results. For the data analysis, it is essential to specify which tool and parameters were used for read mapping. Additionally, how the authors dealt with reads mapping to both homeologous gene pairs needs to be explained. The number of subgenome-specific reads used to measure the differential abundance between homeologous gene pairs should also be provided. And totally how many pairs of homeologous genes were detected in the 0 time point? This section is crucial for understanding how subgenomic asymmetry of homeologous gene expression is measured.

Response:

We thank the Reviewer for the insightful comment.

For RNA-seq experiments, three independent biological replicates were assessed. We performed the poly A selection and conducted the 150 bp pair-end read sequencing on the BGISEQ-500 platform with strand-specific RNA sequencing. Hisat2 v2.1.0⁸ was used for mapping the reads to the durum wheat genome assembly (Svevo RefSeq 1.0)⁹ corrected by accession-specific SNVs information from previous study². Before the mapping, we have assessed the impact of read length on the mapping specificity of A and B homeologous gene pairs (**Rebuttal Fig. 12**). Pairwise sequence alignments were performed on 26,773 homeologous gene pairs, and we simulated the mapping specificity for various

read lengths using a k-mer approach. Our results demonstrated that the 150 bp single-end read length could achieve unique mapping for 95.41% of the reads, whilst the 150 bp paired-end read lengths (equivalent to 300 base pair reads) could yield unique mapping for 99.154% of the reads. Thus, in this work, we used the 150 bp pair-end read sequencing that is sufficient for uniquely assigning the reads to each subgenome. We can confidently detect 12,948 homoeologous gene pairs across all the time points.

Rebuttal Fig. 12 | Impact of read length on mapping specificity of A and B subgenomes.

4. In the legend for Figure 1b, the heatmap shows labels for fast decay and slow decay on the left. Is the heatmap ordered by mRNA decay rate? How is the decay rate defined? Neither the figure legend nor the Decay Profile Normalization method describes this detail clearly. The color bar and figure legend indicate that the values in the heatmap represent the relative transcript abundance over six time points, but do not include the 0-minute point. Are these values the transcript abundance (RPM) or the fold change in RNA abundance at each given time point relative to the 0-minute point? If so, this should be clearly described in the figure legend.

Response:

We thank the Reviewer for the insightful comment.

The heatmap in **Fig. 1b** is ordered by mRNA decay rate. The mRNA decay rate was determined using a mathematical modelling approach based on the maximum likelihood method applied to data obtained with the cordycepin inhibitor method¹⁰. For modelling the decay rate, this analysis referred to the R package RNAdecay (<https://bioconductor.org/packages/release/bioc/html/RNAdecay.html>). Firstly, the lowly expressed genes were filtered with $RPM_{0\ min} < 1$ in 3 replicates, and the other genes were reserved for the later modelling analysis. Then, we normalized the mean value of abundance (RPM) of treated samples by comparison with initial samples at gene level. Subsequently, in order to correct the abundance of the total pool of RNA, the “Decay Factor” was adopted (**Supplementary Data 8**). The genes selected as the decay factor have the following characteristics¹⁰: 1) highly expressed; 2) stable. After rearranging the RNA decay data for modelling, RNA decay rate was modelled using a two model method (both time-dependent exponential decay model and constant exponential decay model were applied) which identified all possibilities of treatment effects on both the decay rate (α) and the decay of the decay rate (β) using maximum likelihood modelling. The decay rate (α) values were applied in our following analysis.

In this study, the foundational equations used for modelling RNA decay rates are as follows¹⁰:

(1) The change in RNA concentration is modelled with a differential equation.

$$\frac{dc}{dt} = -A(t)c$$

The value of $c(t)$ represents the RNA abundance at time t and the value of $A(t)$ denotes the RNA decay rate. After the application of the cordycepin treatment, RNA transcription is inhibited, leading to a decrease in RNA abundance, which is expected to follow a decreasing trend. The initial time point of each transcript is normalized, with $c(0) = 1$.

(2) According to formula (1), the RNA decay rate can be derived.

$$c(t) = e^{-\int A(t)dt}$$

(3) If the decay rate is constant, ($A(t) = \alpha$), then $c(t)$ follows a constant exponential decay model.

$$c(t) = e^{-\alpha t}$$

(4) When the transcript approaches a nonzero value, decay rate is considered to be $A(t) = \alpha e^{-\beta t}$, and $c(t)$ follows a time-dependent exponential decay model.

$$c(t) = e^{-\frac{\alpha}{\beta}(1-e^{-\beta t})}$$

During the modeling process, the “mod_optimization” function in the RNADecay package was used to obtain statistical results. The model with the minimum value of AICc (Akaike information criterion) was determined to be the optimal model among 6 models involved in modeling for each gene.

The values shown in the heatmap represent the fold change in RNA abundance (RPM) at each time point relative to the 0-minute point. At the 0-minute point, the relative decay rate for all genes is set to 1, and the uniform 0-minute values are omitted from the heatmap. We have revised the figure legend to include a clear definition of the decay rate and specify that the heatmap values represent fold changes in RNA abundance.

5. Lines 145-185: The authors investigated the correlation between each mRNA feature and the stability of wheat mRNA individually. A supervised learning approach could be used to deduce the feature importance by using all the features as input for modeling the mRNA decay rate.

Response:

We thank the Reviewer for the comment. We have implemented machine learning approaches for modeling to deduce the feature importance of different features. We inputted the features including the sequence length, GC content and mean SHAPE reactivity of the transcript, 5' UTR, CDS and 3' UTR, as well as the tRNA adaptation index (tAI), totalling 13 features. A total of 13 features were utilized, with the decay rate serving as the label. We divided the dataset into training and testing sets according to an 8:2 ratio. We explored various machine learning regression models, including Random Forest, Gradient Boosting, XGBoost, Extra Trees, Linear Regression, Ridge Regression, Lasso Regression, Elastic Net and K-Nearest Neighbours (**Rebuttal Fig. 13a**). These models were chosen to cover a range of algorithms, from tree-based methods to linear models and ensemble techniques. Among these, the Random Forest Regressor emerged as the best-performing model. However, it only achieved an R^2 value of 0.158 (**Rebuttal Fig. 13b**), which we consider insufficient for robust predictive modelling. Given these results, these features may not be enough for effective machine learning modelling of mRNA decay rates. This may be because the length of mRNA sequences is quite long, and the features, we extracted likely only account for certain fractions of the factors contributing to decay rates.

Rebuttal Fig. 13 | Performance comparison of machine learning models for predicting wheat mRNA decay rates. **a**, The bar plot illustrates the performance of various machine learning regression models in predicting mRNA decay rates. Models tested include Random Forest, Gradient Boosting, XGBoost, Extra Trees, Linear Regression, Ridge Regression, Lasso Regression, Elastic Net, and K-Nearest Neighbours. **b**, The scatter plot displays the R² values for Random Forest.

6. *Why are intronless genes (intron number = 0) less stable? Does the number of introns affect RNA half-life?*

Response:

We thank the Reviewer for the insightful comment. The observation that the intronless genes tend to be less stable in wheat was also observed in yeast¹¹, human¹² and *Arabidopsis*¹³. There are several possible explanations: 1) The components of messenger ribonucleoprotein particles (mRNPs) that are recruited during intron splicing or deposited onto exon-exon junctions may be retained in cytoplasmic mRNPs. These components could potentially serve as signals of mRNA stability or as insulators, preventing inter- or intra-RNA base-pairing. Some proteins have demonstrated insulating effects to hinder RNA: DNA hybrid formation in the nucleus, thereby suppressing undesired DNA recombination. Moreover, it is conceivable that genes with a high intron density would produce mRNAs that are better insulated in the cytoplasm. 2) Human genes lacking introns or containing few introns are prone to forming stem-loop structures in their mRNAs, leading to ribosome stalling and subsequent endonucleolytic mRNA cleavage. Additionally, these mRNAs have an increased tendency to engage in double-stranded RNA interactions with other cytoplasmic RNAs, rendering them susceptible to degradation through RNA interference. 3) Evidence suggests that the exon junction complexes within mRNPs facilitate the association of mRNA with ribosomes. The ribosomes bound to mRNA may contribute to mRNA stabilization, akin to the protein constituents of mRNPs.

Furthermore, we have calculated the average half-life for mRNAs with different numbers of introns. Overall, we found that genes without introns exhibit the shortest half-lives in comparison to genes containing multiple introns (**Rebuttal Fig. 14**). There is a trend of increasing half-lives as the number of introns increases up to six introns (**Rebuttal Fig. 14**). For genes with more than six introns, the trend is not obvious (**Rebuttal Fig. 14**). This may be due to the relatively small number of genes that contain more than six introns.

Rebuttal Fig. 14 | Boxplot of mRNA average half-lives across genes with varying numbers of introns.

7. Lines 246-255: Two accessions (how many replicates?) were selected for the RNA-seq experiment (Fig. 5a). However, these RNA-seq libraries were not described in the methods section, nor are they mentioned in the data availability statement.

Response:

We thank the Reviewer for the comment. For each accession, we performed three biological replicates for the RNA-seq experiment. We have provided all the details of these RNA-seq libraries in the methods section along with the detailed clarification in the data availability statement.

8. Lines 275-286: One EMS mutant with a C to U mutation in the 3' UTR of TRITD4Av1G006890 was detected in a Kronos EMS library. However, no references related to this EMS library were provided. If this library has not been previously published, details about the Kronos EMS library should be described. Importantly, the method for detecting the mutant with the C to U mutation is also missing.

Response:

We thank the Reviewer for the comment.

The Kronos EMS library was published by the Dubcovsky Lab in 2017, as part of a study that aimed to induce and catalogue mutations in tetraploid and hexaploid wheat¹⁴. Specifically, this study identified 4.15 million uniquely mapped EMS-type mutations (G to A and C to T) in tetraploid wheat¹⁴. The method used for detecting the mutations involved sequencing the protein-coding regions of 1,535 mutant lines from the tetraploid wheat variety "Kronos" using a wheat exome capture platform¹⁴. A widely used computational pipeline, Mutation and Polymorphism Survey (MAPS), was applied to identify the high the mutations in both tetraploid and hexaploid wheat¹⁵. To ensure the high confidence of identified mutations, the researchers also optimised the parameters locally based on their own data to minimize the detection of false mutations without losing too much sensitivity for both heterozygous and homozygous mutations¹⁴. Approximately 90% of the captured wheat genes were analyzed, with deleterious alleles being identified. Researchers are able to access this database online (dubcovskylab.ucdavis.edu/wheat_blast and www.wheat-tilling.com), where they can search for

specific mutations in their target genes and request seeds to study gene function or improve wheat varieties. We have edited this part in our method section in our revised manuscript.

References

1. Yang X., et al. Wheat in vivo RNA structure landscape reveals a prevalent role of RNA structure in modulating translational subgenome expression asymmetry. *Genome Biol.* **22**, 326 (2021).
2. Zhou Y., et al. *Triticum* population sequencing provides insights into wheat adaptation. *Nat. Genet.* **52**, 1412-1422 (2020).
3. McCaskill J. S. The equilibrium partition function and base pair binding probabilities for RNA secondary structure. *Biopolymers* **29**, 1105-1119 (1990).
4. Ronny Lorenz, et al. ViennaRNA Package 2.0. *Algorithms Mol. Biol.* **6**, 1-14 (2011).
5. Rang F. J., Kloosterman W. P. & de Ridder J. From squiggle to basepair: computational approaches for improving nanopore sequencing read accuracy. *Genome Biol.* **19**, 90 (2018).
6. Lima L., et al. Comparative assessment of long-read error correction software applied to Nanopore RNA-sequencing data. *Brief. Bioinformatics* **21**, 1164-1181 (2020).
7. Shi X. & Ling H.-Q. Current advances in genome sequencing of common wheat and its ancestral species. *Crop J.* **6**, 15-21 (2018).
8. Kim D., Langmead B. & Salzberg S. L. HISAT: a fast spliced aligner with low memory requirements. *Nat. Methods* **12**, 357-360 (2015).
9. Maccaferri M., et al. Durum wheat genome highlights past domestication signatures and future improvement targets. *Nat. Genet.* **51**, 885-895 (2019).
10. Sorenson R. S., Deshotel M. J., Johnson K., Adler F. R. & Sieburth L. E. *Arabidopsis* mRNA decay landscape arises from specialized RNA decay substrates, decapping-mediated feedback, and redundancy. *Proc. Natl. Acad. Sci. U.S.A.* **115**, E1485-E1494 (2018).
11. Wang Y., et al. Precision and functional specificity in mRNA decay. *Proc. Natl. Acad. Sci. U.S.A.* **99**, 5860-5865 (2002).
12. Yang E., et al. Decay rates of human mRNAs: correlation with functional characteristics and sequence attributes. *Genome Res.* **13**, 1863-1872 (2003).
13. Narsai R., et al. Genome-wide analysis of mRNA decay rates and their determinants in *Arabidopsis thaliana*. *Plant Cell* **19**, 3418-3436 (2007).
14. Krasileva K. V., et al. Uncovering hidden variation in polyploid wheat. *Proc. Natl. Acad. Sci. U.S.A.* **114**, E913-E921 (2017).
15. Henry I. M., et al. Efficient genome-wide detection and cataloging of EMS-induced mutations using exome capture and next-generation sequencing. *Plant Cell* **26**, 1382-1397 (2014).

REVIEWERS' COMMENTS

Reviewer #1 (Remarks to the Author):

I would like to express my sincere gratitude for the efforts of the authors. All of my concerns have been thoroughly addressed, and I have no more questions.

Reviewer #2 (Remarks to the Author):

The authors have addressed all of my comments. I am satisfied with the current version.

Reviewer #3 (Remarks to the Author):

The authors have addressed all my concerns well, and I enjoy reading the revised manuscript. I have no more concerns.

Reviewer #4 (Remarks to the Author):

The authors have addressed all of my comments.